# Distinct roles of ATM and ATR in the regulation of ARP8 phosphorylation to prevent chromosome translocations

Jiying Sun[1], Lin Shi[1], Aiko Kinomura[1], Atsuhiko Fukuto[1,2], Yasunori Horikoshi[1], Yukako Oma[3], Masahiko Harata[3], Masae Ikura[4], Tsuyoshi Ikura[4], Roland Kanaar[5], Satoshi Tashiro[1]*

[1]Department of Cellular Biology, Research Institute for Radiation Biology and Medicine, Hiroshima University, Hiroshima, Japan; [2]Department of Ophthalmology and Visual Science, Graduate School of Biomedical Sciences, Hiroshima University, Hiroshima, Japan; [3]Laboratory of Molecular Biology, Graduate School of Agricultural Science, Tohoku University, Sendai, Japan; [4]Laboratory of Chromatin Regulatory Network, Department of Mutagenesis, Radiation Biology Center, Kyoto University, Kyoto, Japan; [5]Department of Molecular Genetics, Oncode Institute, Rotterdam, Netherlands

**Abstract** Chromosomal translocations are hallmarks of various types of cancers and leukemias. However, the molecular mechanisms of chromosome translocations remain largely unknown. The ataxia-telangiectasia mutated (ATM) protein, a DNA damage signaling regulator, facilitates DNA repair to prevent chromosome abnormalities. Previously, we showed that ATM deficiency led to the 11q23 chromosome translocation, the most frequent chromosome abnormalities in secondary leukemia. Here, we show that ARP8, a subunit of the INO80 chromatin remodeling complex, is phosphorylated after etoposide treatment. The etoposide-induced phosphorylation of ARP8 is regulated by ATM and ATR, and attenuates its interaction with INO80. The ATM-regulated phosphorylation of ARP8 reduces the excessive loading of INO80 and RAD51 onto the breakpoint cluster region. These findings suggest that the phosphorylation of ARP8, regulated by ATM, plays an important role in maintaining the fidelity of DNA repair to prevent the etoposide-induced 11q23 abnormalities.
DOI: https://doi.org/10.7554/eLife.32222.001

*For correspondence:
ktashiro@hiroshima-u.ac.jp

Competing interests: The authors declare that no competing interests exist.

## Introduction

Chromosome translocations are one of the most common types of genetic rearrangements induced by DNA damaging agents, such as ionizing radiation and certain chemotherapies. The presence of disease-specific chromosome translocations, especially in hematological malignancies such as the t (9;22) or Philadelphia chromosome in chronic myelocytic leukemia, has been reported. Molecular studies of the breakpoints of such disease-specific chromosome translocations have revealed the clustering of the breakpoints in specific regions, designated as the breakpoint cluster region (BCR). In lymphoid malignancies, the involvement of the physiological recombination of immunoglobulin and T-cell receptor genes in chromosome translocations has been suggested, due to the presence of signal sequences for the recombination at the breakpoints. However, the molecular mechanisms of chromosome translocations in other cell types remain largely unknown.

Chromosome translocations arise as a consequence of errors in the repair of DNA double strand breaks (DSBs). Eukaryotic cells utilize a variety of repair pathways for DSBs, including two major ones, non-homologous end joining (NHEJ) and homologous recombination repair (HR). In the

absence of these canonical pathways, the activation of the alternative NHEJ (Alt-EJ) pathway and the inactivation of DNA polymerase theta are implicated in chromosomal translocations (*Zelensky et al., 2017*) (*Mizuno et al., 2009*; *Ruiz et al., 2009*; *Schmidt et al., 2006*). In contrast, HR is regarded as a precise DSB repair system, since either the intact sister chromatid or the homologous region is used as the template for repair. However, both the depletion and overexpression of the RAD51 recombinase, a key factor involved in HR, lead to chromosomal abnormalities (*Reliene et al., 2007*; *Richardson et al., 2004*). Therefore, the precise regulation of the recombination activity is also required for DNA repair to prevent chromosome translocations.

DNA damage leads to the activation of the DNA damage response and repair pathways. The ataxia-telangiectasia mutated (ATM) protein regulates the DNA damage response in reaction to DSBs, through its kinase activity (*Clouaire et al., 2017*; *Guleria and Chandna, 2016*; *Shiloh, 2003*; *Shiloh and Ziv, 2013*). Alterations in the function of ATM play pathologic roles in the development of leukemia/lymphoma and cancer (*Khanna, 2000*; *Oguchi et al., 2003*; *Reliene et al., 2007*). Chromosome translocations involving the MLL gene on 11q23 are the most frequent chromosome abnormalities in secondary leukemia associated with chemotherapy employing etoposide, a topoisomerase II poison. An increase of 11q23 translocations is observed in the ATM kinase activity-deficient fibroblast cell line AT5BIVA (*Nakada et al., 2006*). We showed previously that a deficiency of ATM, a DNA damage signaling kinase, led to the excessive binding of RAD51 and the chromatin remodeling factor INO80 to the BCR in the MLL gene after etoposide treatment (*Sun et al., 2010*). INO80 is conserved in eukaryotes and acts as an integral scaffold for assembling other proteins into the INO80 chromatin remodeling complex (*Chen et al., 2011*; *Morrison et al., 2004*). The INO80 complex plays an important role in chromatin reorganization for transcription (*Lafon et al., 2015*; *Xue et al., 2015*), replication (*Falbo and Shen, 2012*; *Vassileva et al., 2014*) and DNA repair (*Alatwi and Downs, 2015*; *Gospodinov et al., 2011*; *Morrison et al., 2004*; *Seeber et al., 2013*; *van Attikum et al., 2004*). The INO80 complex is required for effective DNA end resection at the early stage of HR in budding yeast and human cells (*Gospodinov et al., 2011*; *Lademann et al., 2017*; *Tsukuda et al., 2005*; *Tsukuda et al., 2009*; *van Attikum et al., 2004*). Therefore, we speculated that the decreased fidelity of DNA repair by the inappropriate regulation of HR repair in ATM-deficient cells could lead to chromosomal translocations. However, the mechanism by which ATM deficiencies induce the excessive binding of INO80 and RAD51 to the BCR after etoposide treatment remains to be clarified.

INO80 forms a chromatin remodeling complex with more than 15 subunits (*Jin et al., 2005*; *Shen et al., 2000*). Among the subunits of the INO80 complex, ARP8 functions as a nucleosome recognition module and enhances the nucleosome-binding affinity of the protein complex (*Saravanan et al., 2012*; *Shen et al., 2003*). In this study, we found that ARP8 is required for the binding of INO80 and RAD51 to the BCR in the MLL gene after etoposide treatment. We also showed that the DNA damage-dependent phosphorylation of ARP8 is regulated by ATM and ATR after etoposide treatment. The ATM-dependent phosphorylation of ARP8 negatively regulates the interaction between INO80 and ARP8, leading to the reduced binding of INO80 and RAD51 to the BCR after etoposide treatment. In contrast, ATR was not involved in the regulation of the etoposide-induced binding of RAD51 to the BCR. These findings suggest that ATM plays distinct roles from ATR in the phosphorylation of ARP8 to prevent the etoposide-induced 11q23 chromosome translocations, through the negative regulation of INO80 and RAD51 binding to the BCR.

## Results

### Phosphorylation of ARP8 after etoposide treatment regulated by ATM and ATR

In our previous study, we showed that ATM deficiency resulted in the overloading of INO80 and RAD51 onto the BCR of the MLL gene after etoposide treatment (*Sun et al., 2010*). The finding led us to investigate the phosphorylation target of the ATM kinase, which could regulate the loading of INO80 and RAD51 onto the BCR. First, we examined the phosphorylation status of INO80 after etoposide treatment. However, we could not detect the phosphorylation of INO80 by ATM after etoposide treatment by an immunoblotting analysis (*Figure 1—figure supplement 1A*). Therefore, we

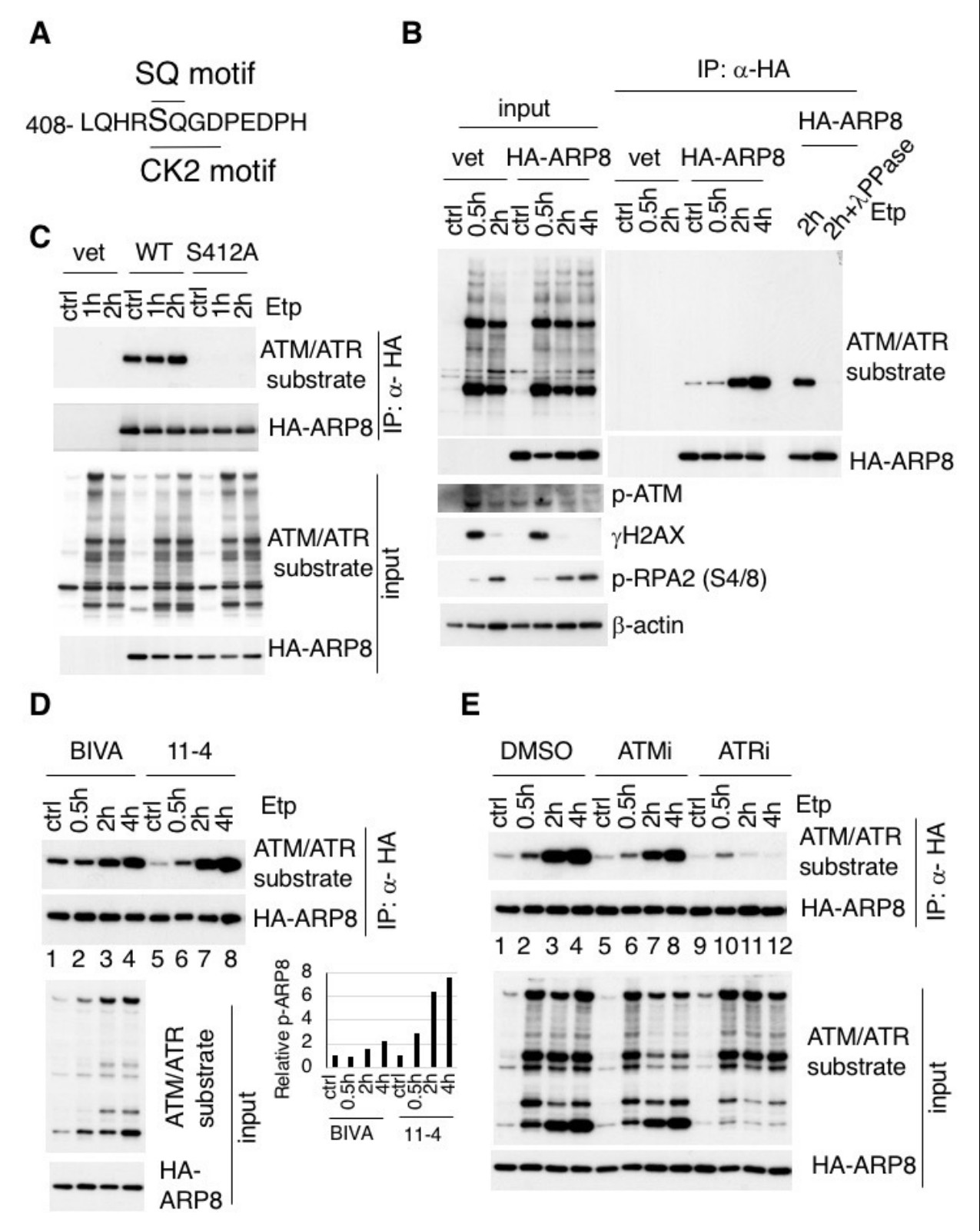

**Figure 1.** Identification of etoposide-induced ARP8 phosphorylation and the possible responsible kinase. (**A**) Amino acid sequence 408 through 420 of ARP8. The Ser412 residue, within the ATM/ATR substrate motif and the CK2 substrate motif, is indicated. (**B**) Immunoprecipitation analysis of ARP8 phosphorylation. U2OS cells transiently expressing an empty HA vector or a vector encoding HA-tagged ARP8 were treated with DMSO (ctrl) or etoposide (Etp) for 15 min, then washed twice and cultured in complete medium for the indicated times. The nuclear extracts were incubated with anti-

*Figure 1 continued on next page*

*Figure 1 continued*

HA-conjugated anti-mouse IgG Dynabeads. The precipitates were electrophoresed through a gel and probed by western blotting with an anti-ATM/ATR substrate antibody or an anti-HA antibody. λPPase treatment identified the band of phosphorylated HA-ARP8. The blot of input was probed by antibodies against phospho-ATM (p-ATM), γ H2AX or phospho-RPA2 at Ser4/8 (p-RPA2). β-actin was used as a loading control. (C) Identification of the ARP8 phosphorylation site by an immunoprecipitation analysis. U2OS cells were transfected with an empty HA vector (vet), or a vector encoding HA-tagged wild-type ARP8 (WT) or HA-ARP8 S412A (S412A) for 48 hr. The cells were washed after treatment with etoposide or DMSO for 15 min, cultured in fresh medium, and harvested at the indicated time points. Whole cell extracts were used for the immunoprecipitation analysis. (D) Etoposide-induced ARP8 phosphorylation in ATM-deficient BIVA and ATM-proficient 11–4 cells. Immunoprecipitation analysis of cell extracts of BIVA or 11–4 cells transfected with HA-tagged wild-type ARP8 using anti-HA antibodies. The cells were treated with DMSO (ctrl) or etoposide (Etp) for 15 min, cultured in fresh medium, and harvested at the indicated time points. Whole cell extracts were used for the immunoprecipitation analysis, which was performed as described in (B). The amounts of phosphorylated ARP8 and HA-ARP8 were quantified, using the Image J software. The results of the quantitative analysis are shown as the relative values to the DMSO controls. Source data are presented in *Figure 1—source data 1*. (E) Immunoprecipitation analysis of cell extracts from 11 to 4 cells expressing HA-tagged ARP8. The cells were treated with DMSO, 10 μM ATMi (KU55933), or 10 μM ATRi (VE821) for 2 hr before etoposide treatment, and then the inhibitors (5 μM) were added after the cells were washed.

DOI: https://doi.org/10.7554/eLife.32222.002

The following source data and figure supplements are available for figure 1:

**Source data 1.** Source raw data for *Figure 1D*.

DOI: https://doi.org/10.7554/eLife.32222.005

**Figure supplement 1.** ARP8 contains an ATM/ATR substrate SQ motif at Ser412 and Q413.

DOI: https://doi.org/10.7554/eLife.32222.003

**Figure supplement 2.** ATM, but not CK2 is responsible for ARP8 phosphorylation after etoposide treatment.

DOI: https://doi.org/10.7554/eLife.32222.004

decided to examine the phosphorylation of the subunits of the INO80 protein complex after etoposide treatment.

The substrates of ATM contain the core sequence with an SQ or TQ motif (*Kim et al., 1999*; *Matsuoka et al., 2007*; *O'Neill et al., 2000*). By searching for SQ or TQ motifs in the subunits of the INO80 complex, we found that APR8 had an SQ motif at S412 and Q413. This motif in ARP8 is indicated as a putative phosphorylation site in the PhosphoSitePlus database (*Figure 1A* and *Figure 1—figure supplement 1B–D*). ARP8 is required for DNA binding by INO80 in yeast and mammals, and the ARP8 knockout in human cells impairs the binding of INO80 to chromatin and causes defects in DNA repair (*Kashiwaba et al., 2010*; *Saravanan et al., 2012*). Therefore, we examined whether ARP8 was the phosphorylation target of ATM. The immunoblotting analysis, using antibodies against the ATM/ATR substrate, revealed that the level of ARP8 phosphorylation was significantly increased from 2 hr after etoposide treatment (*Figure 1B*). The disappearance of the signal by a protein phosphatase treatment validated that the derived signal resulted from the phosphorylation of ARP8 (*Figure 1B*).

To confirm that Ser412 is the site in ARP8 that is phosphorylated after etoposide treatment, we produced an ARP8 mutant by replacing Ser412 with alanine. An immunoprecipitation analysis revealed that the etoposide-induced phosphorylation was completely abolished by the S412A substitution, indicating that Ser412 is the sole site within ARP8 that is phosphorylated in response to etoposide treatment (*Figure 1C*).

Next to investigate whether the etoposide-induced phosphorylation of ARP8 is regulated by ATM, we compared the phosphorylation status of ARP8 in ATM-deficient BIVA and ATM-proficient 11–4 cells after etoposide treatment. An immunoblotting analysis revealed that the etoposide-induced ARP8 phosphorylation in BIVA cells was lower than that in 11–4 cells (*Figure 1D*). Moreover, the phosphorylation of ARP8 in ATM-proficient 11–4 cells after etoposide treatment was repressed by the ATM-specific inhibitor, KU55933 (ATMi) (*Figure 1E*; lanes 1–8). These findings suggest that ATM regulates the ARP8 phosphorylation after etoposide treatment. The dose-dependent repression of the etoposide-induced ARP8 phosphorylation by ATMi in U2OS cells further supported this notion (*Figure 1—figure supplement 2A*).

Since ATMi did not completely abolish the etoposide-induced phosphorylation of ARP8, we next examined the relevance of ATR, another PI3-family kinase member sharing the same phosphorylation motif with ATM. ATR is responsible for the phosphorylation in the DNA replication stress response, and is activated by ATM after ionizing radiation (*Cuadrado et al., 2006*; *Jazayeri et al., 2006*; *Myers and Cortez, 2006*). The etoposide-induced phosphorylation of ARP8 was strongly

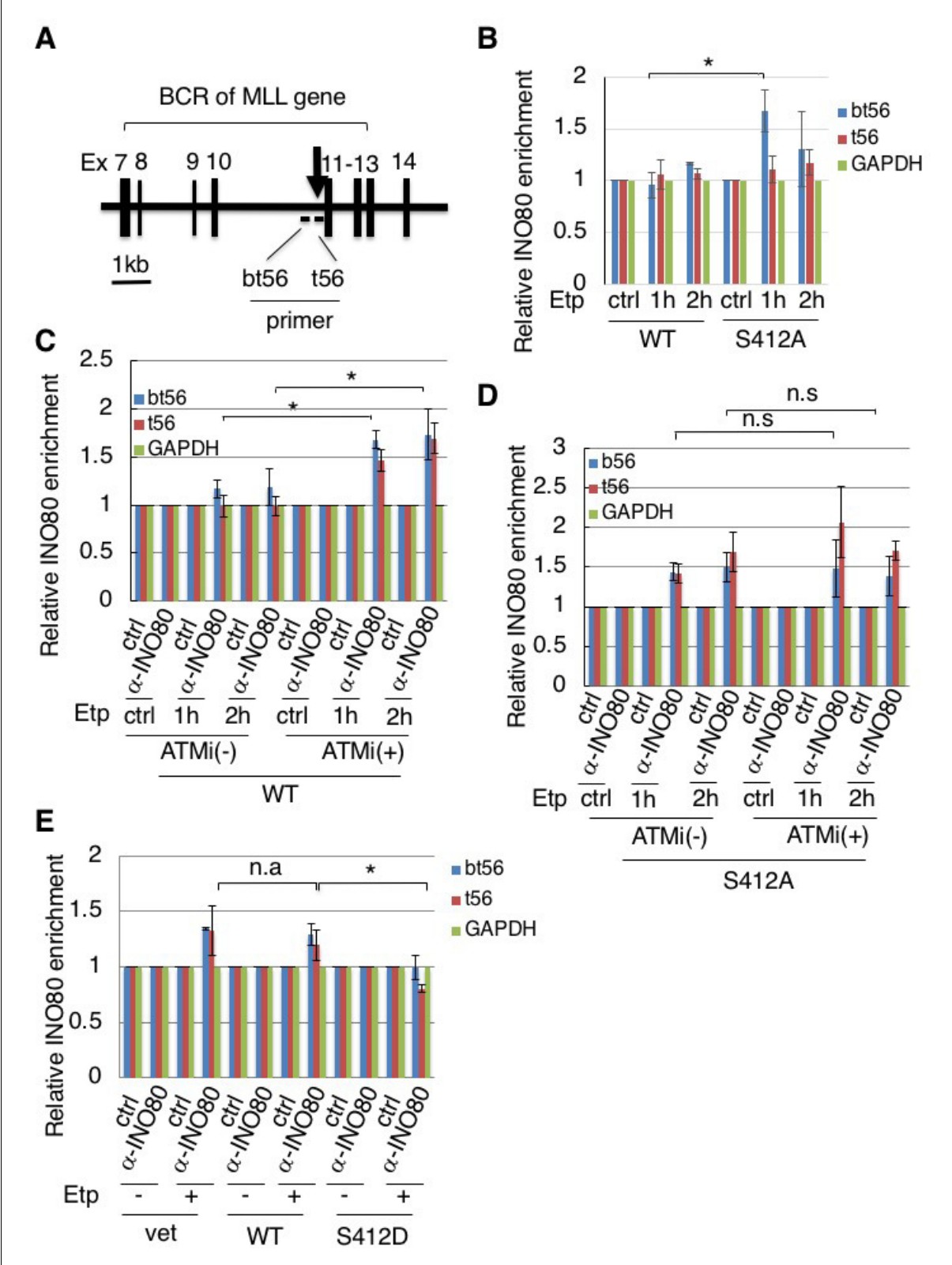

**Figure 2.** Phosphorylation of ARP8 negatively regulates the etoposide-induced enrichment of INO80. (**A**) Schematic representation of the BCR in the MLL gene. The locations of the primers used in the real-time PCR analyses are shown. The arrow indicates 11q23 chromosome translocation breakpoint hotspot identified in treatment-related leukemia. Ex: Exon. (**B**) ChIP analysis of the INO80 loading onto the MLL BCR in endogenous ARP8-depleted11-4 Flp-In cells expressing either the siRNA-resistant wild-type (WT) or phospho-deficient ARP8 (S412A) after tetracycline treatment. The cells were

*Figure 2 continued on next page*

*Figure 2 continued*

treated with DMSO (ctrl) or etoposide for 15 min, washed, and then cultured in fresh medium for 1 or 2 hr. GAPDH is shown as the control region. Values represent the means ± SE from three independent experiments. *: p<0.05. Source data are presented in *Figure 2—source data 1*. (C) ChIP analysis of the INO80 loading onto the MLL BCR in wild-type ARP8 expressing11-4 Flp-In cells. The cells were treated with/without an ATM inhibitor (KU55933) for 2 hr before etoposide treatment, and then the inhibitors (5 µM) were added after the cells were washed. The experiment was performed as described in (B). Values represent the means ± SE from three independent experiments. *: p<0.05. The level of ATM phosphorylation or expression of INO80 was shown in *Figure 2—figure supplement 2B*. Source data are presented in *Figure 2—source data 1*. (D) ChIP analysis of the INO80 loading onto the MLL BCR in S412A ARP8 expressing 11–4 Flp-In cells. The cells were treated with/without 10 µM ATM inhibitor for 2 hr before etoposide treatment, and then the inhibitors (5 µM) were added after the cells were washed. The experiment was performed as described in (B). Values represent the means ± SE from three independent experiments. n.s: no significant difference. The levels of ATM phosphorylation and INO80 expression are shown in *Figure 2—figure supplement 2C*. Source data are presented in *Figure 2—source data 1*. (E) ChIP analysis of the INO80 loading onto the MLL BCR in endogenous ARP8-depleted BIVA cells transfected with either the siRNA-resistant wild-type (WT) or phospho-mimetic ARP8(S412D). The control cells were transfected with an empty vector and a non-targeting siRNA (vet). The cells were treated with DMSO (ctrl) or etoposide for 15 min, washed, and then cultured in fresh medium for 1 hr. Values represent the means ± SE from three independent experiments. *p<0.05, n.a: not analyzed. Source data are presented in *Figure 2—source data 1*.

DOI: https://doi.org/10.7554/eLife.32222.006

The following source data and figure supplements are available for figure 2:

**Source data 1.** Source raw data for *Figure 2B-E*.
DOI: https://doi.org/10.7554/eLife.32222.009
**Source data 2.** Source raw data for *Figure 2—figure supplement 1A and B*.
DOI: https://doi.org/10.7554/eLife.32222.010
**Figure supplement 1.**
DOI: https://doi.org/10.7554/eLife.32222.007
**Figure supplement 2.** Establishment of stable and inducible ARP8 expressing 11–4 Flp-In cells.
DOI: https://doi.org/10.7554/eLife.32222.008

repressed by the ATR inhibitor VE821 (ATRi) in 11–4 cells (*Figure 1E*, lanes 9–12). This finding suggests that ATR is the major kinase responsible for etoposide-induced ARP8 phosphorylation. In contrast, casein kinase 2, another kinase involved in the DNA damage response (*Olsen et al., 2012*) (*Guerra et al., 2014*) was not involved in the phosphorylation of ARP8 after etoposide treatment (*Figure 1—figure supplement 2B*). Together, these findings suggest that the phosphorylation of ARP8 at S412 after etoposide treatment is regulated by ATM and ATR.

## Negative regulation of the etoposide-induced loading of INO80 onto the MLL BCR by the phosphorylation of ARP8

Having established that ARP8 is phosphorylated after etoposide treatment, we investigated the role of ARP8 in INO80 loading onto the MLL BCR (*Figure 2A*). The enrichment of γH2AX on BCR was observed in ATM-proficient 11–4 cells after etoposide treatment, suggesting that etoposide induces DNA damage specifically at the BCR (*Figure 2—figure supplement 1A*). A chromatin immunoprecipitation (ChIP) analysis revealed that the depletion of ARP8 reduced the binding of INO80 to the BCR after etoposide treatment in BIVA cells ((*Figure 2—figure supplement 1B*). Since the depletion of ARP8 did not affect the levels of INO80 (*Figure 2—figure supplement 1C*), the result suggests that ARP8 is required for loading INO80 onto the MLL BCR after etoposide treatment in BIVA cells. Next, to explore the role of ARP8 phosphorylation in the regulation of INO80 loading onto the BCR after etoposide treatment, we generated ATM-proficient 11–4 cell lines expressing the siRNA-resistant wild-type ARP8 (WT) or the phosphorylation-deficient mutant S412A (*Figure 2—figure supplement 2A*). The ChIP analysis revealed that the expression of the S412A mutant in ATM-proficient cells increased the binding of INO80 to the BCR after etoposide treatment (*Figure 2B*). Moreover, ATMi treatment increased the binding of INO80 to the BCR after etoposide treatment in ATM-proficient cells expressing wild-type ARP8 (*Figure 2C*). These findings suggest that the etoposide-induced phosphorylation of ARP8 represses the binding of INO80 to the BCR. Importantly, the ATMi treatment failed to enhance the enrichment of INO80 at the BCR in ATM-proficient 11–4 cells expressing the ARP8 S412A mutant (*Figure 2D*). This suggests that ARP8 phosphorylation at S412 regulated by ATM is responsible for the excessive binding of INO80 at the BCR after etoposide treatment. To further confirm the repression of the INO80 binding to the BCR by the ARP8 phosphorylation, we next introduced the siRNA-resistant phosphomimetic mutant S412D into the ATM-

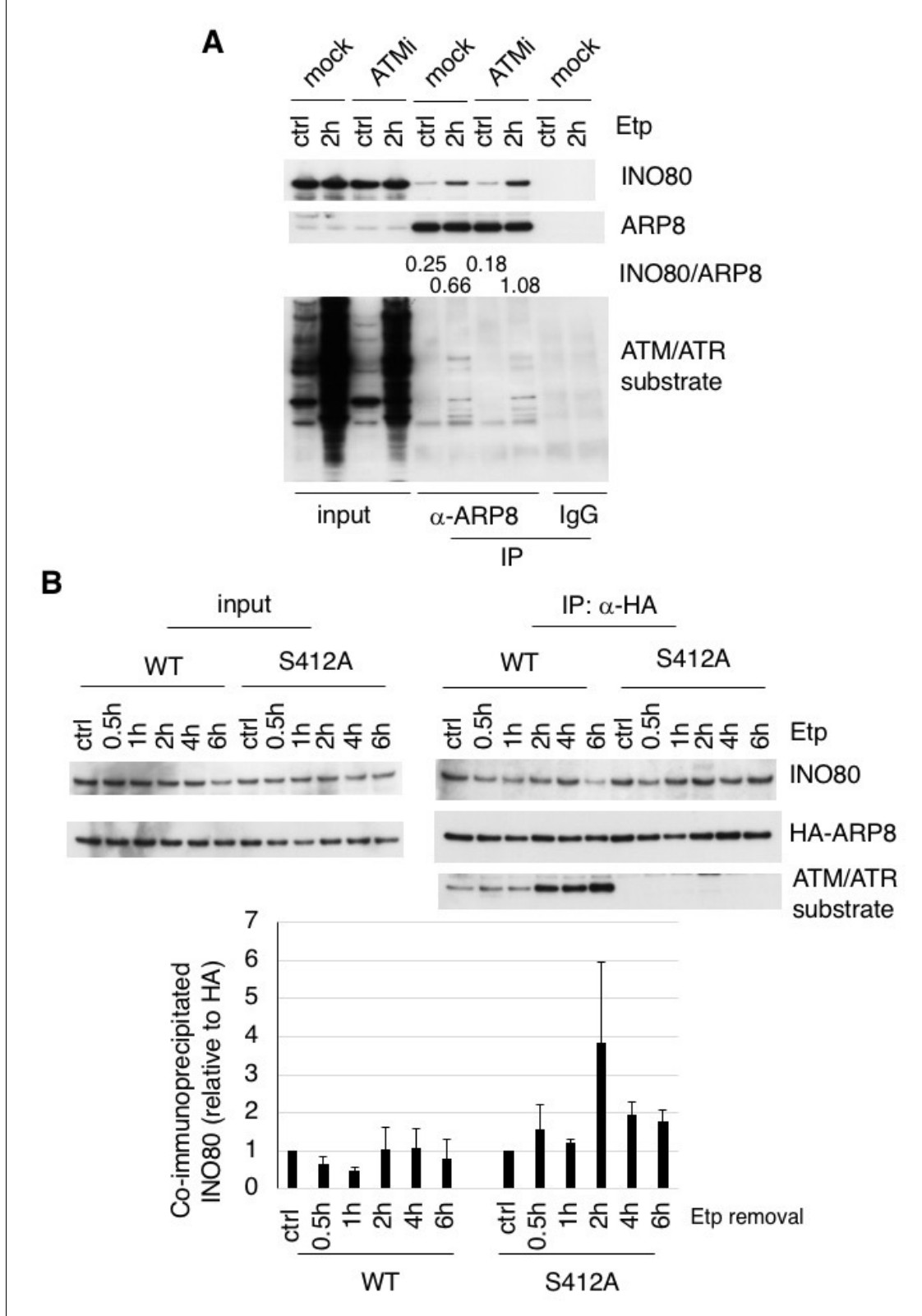

**Figure 3.** ARP8 phosphorylation deficiency increased its interaction with INO80. (**A**) Immunoprecipitation analysis of the interaction between INO80 and ARP8 in ATM inhibitor treated U2OS cells. The cells were treated with 10 μM KU55933 (ATMi) or equal amounts of DMSO (mock) for 2 hr, and then treated with DMSO (ctrl) or etoposide for 15 min, washed, and then cultured in fresh medium with or without 5 μM KU55933 for 2 hr. Immunoprecipitation analysis was performed with either anti-ARP8 antibodies or normal IgG. The relative immunoprecipitated amounts of INO80 are

*Figure 3 continued on next page*

*Figure 3 continued*

shown. Quantitative analysis was performed using the Image J software. (B) Examination of the interaction between INO80 and ARP8 in U2OS cells expressing HA-tagged wild-type or S412A ARP8. The endogenous ARP8-depleted cells were treated with etoposide for 15 min. After the cells were washed, they were placed in fresh medium and harvested at the indicated time points. The nuclear extracts were incubated with anti-HA-conjugated anti-mouse IgG Dynabeads. The precipitates were electrophoresed through a gel and probed by western blotting with an anti-INO80 or an anti-HA or an anti-ATM/ATR substrate antibody. The amounts of INO80 and HA-ARP8 were quantified, using the Image J software. The results of quantitative analysis are shown as the relative values as compared to the DMSO control from three independent experiments. Source data are presented in *Figure 3—source data 1*.
DOI: https://doi.org/10.7554/eLife.32222.011
The following source data and figure supplements are available for figure 3:
**Source data 1.** Source raw data for *Figure 3B*.
DOI: https://doi.org/10.7554/eLife.32222.014
**Source data 2.** Source raw data for *Figure 3—figure supplement 1B*.
DOI: https://doi.org/10.7554/eLife.32222.015
**Source data 3.** Source raw data for *Figure 3—figure supplement 2B*.
DOI: https://doi.org/10.7554/eLife.32222.016
**Figure supplement 1.** Examination of the interaction between ARP8 and INO80.
DOI: https://doi.org/10.7554/eLife.32222.012
**Figure supplement 2.** Examination of ARP8-INO80 interactions using Proximity ligation assay.
DOI: https://doi.org/10.7554/eLife.32222.013

deficient BIVA cells. Consistently, the expression of the phosphomimetic mutant S412D in BIVA cells reduced the binding of INO80 to the BCR after etoposide treatment (*Figure 2E*). Taken together, these results indicate that the phosphorylation of ARP8 represses the loading of INO80 onto the MLL BCR in response to etoposide-induced damage.

## Phosphorylation of ARP8 regulates its interaction with INO80

To determine whether the phosphorylation of ARP8 affects the loading of INO80 onto the MLL BCR through its interaction with INO80, we performed an immunoprecipitation analysis using U2OS cells. We found that etoposide treatment increased the interaction between INO80 and ARP8 in the ATM-proficient U2OS cells (*Figure 3A*). The etoposide-induced interaction between INO80 and ARP8 was enhanced by the treatment of U2OS cells with an ATM inhibitor (*Figure 3A*). These findings raised the possibility that ATM negatively regulates the interaction of ARP8 with INO80 after etoposide treatment. Therefore, we investigated the role of the ATM-regulated phosphorylation of ARP8 in the interaction between ARP8 and INO80 after etoposide treatment by using the U2OS cells stably expressing the phosphorylation-deficient ARP8 S412A mutant. An immunoprecipitation analysis showed an increased level of interaction of the ARP8 S412A mutant with INO80 after etoposide treatment, as compared to that of wild-type ARP8 (ARP8 WT) in ATM-proficient cells (*Figure 3B*). Similar results were obtained using the U2OS cells transiently expressing the ARP8 S412A mutant (*Figure 3—figure supplement 1A*). These findings support the notion that the phosphorylation of ARP8 represses its interaction with INO80 after etoposide treatment. To confirm the repression of the association of ARP8 with INO80 by ATM, we transiently expressed HA-tagged ARP8 WT or S412D mutant in ATM-deficient BIVA cells. An immunoprecipitation analysis revealed the increased interaction of INO80 with ARP8 WT, but not with the S412D mutant, in ATM-deficient BIVA cells after etoposide treatment (*Figure 3—figure supplement 1B*). A proximity ligation assay confirmed the decreased interaction of INO80 with the ARP8 S412D mutant (*Figure 3—figure supplement 2*). Taken together, these findings suggest that the phosphorylation of ARP8 represses the interaction between INO80 and ARP8 after etoposide treatment.

## ARP8 phosphorylation negatively regulates RAD51 loading onto the BCR after etoposide treatment

In our previous study, we detected the excess loading of RAD51 onto the BCR of the MLL gene in etoposide-treated ATM-deficient BIVA cells (*Sun et al., 2010*). In yeast, INO80 promotes DNA end resection and RAD51 binding to ssDNA for homology search/invasion during HR (*Lademann et al., 2017*; *Tsukuda et al., 2009*). To explore the mechanism of excessive RAD51 loading on the BCR,

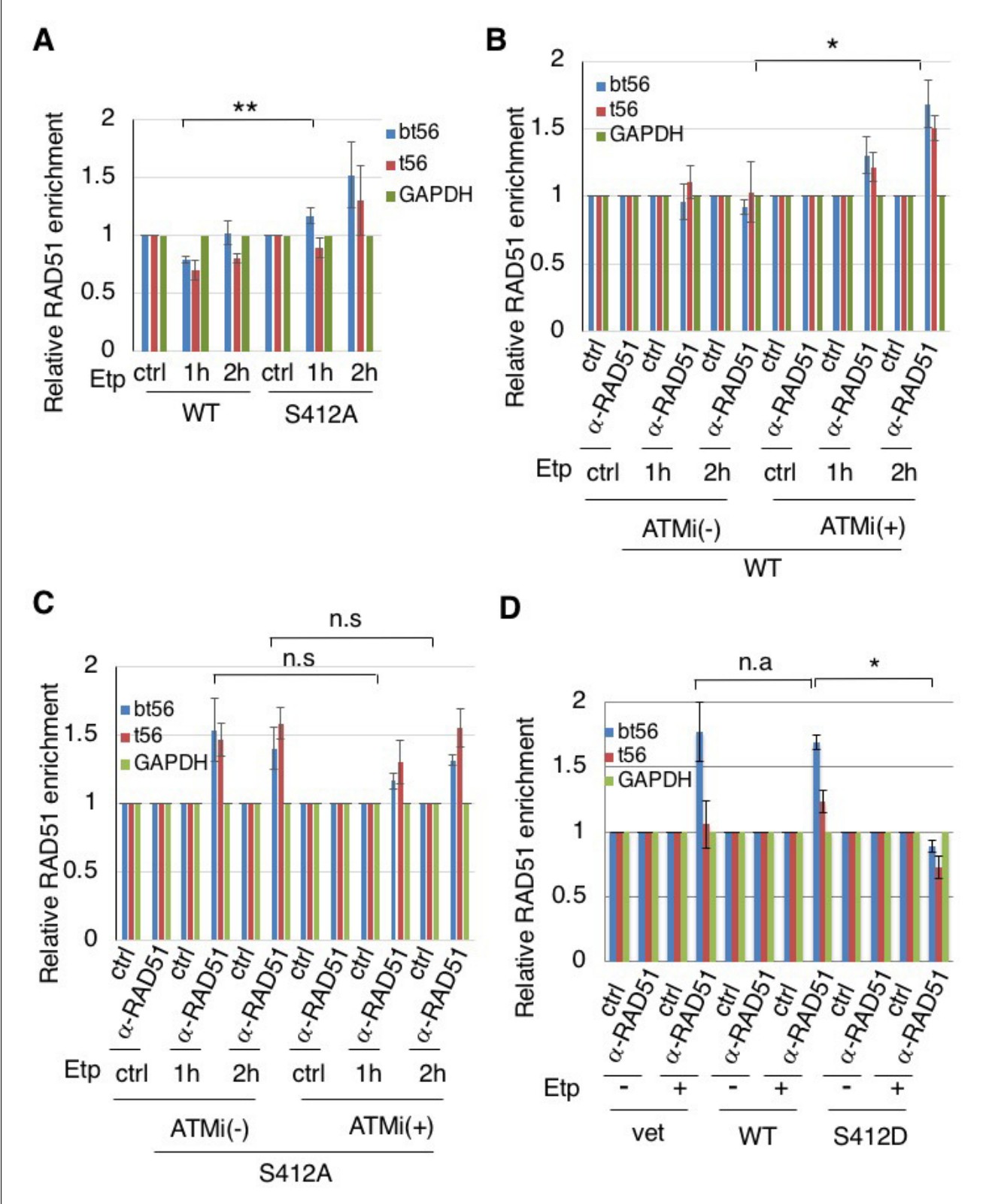

**Figure 4.** ARP8 phosphorylation prevents the excessive RAD51 loading onto MLL BCR. (**A**) ChIP analysis of the RAD51 loading onto the MLL BCR in endogenous ARP8-depleted11-4 Flp-In cells expressing the siRNA-resistant wild-type (WT) or phospho-deficient ARP8 (S412A) after tetracycline treatment. The cells were treated with DMSO (ctrl) or etoposide for 15 min, washed, and then cultured in fresh medium for 1 or 2 hr. Values represent the means ± SE from three independent experiments. **: p<0.01. Source data are presented in *Figure 4—source data 1*. (**B**) ChIP analysis of the

*Figure 4 continued on next page*

*Figure 4 continued*

RAD51 loading onto the MLL BCR in wild-type ARP8 expressing 11–4 cells. Following a treatment with/without 10 µM ATM inhibitor (KU55933) for 2 hr, the cells were treated with DMSO (ctrl) or etoposide for 15 min, washed, and then cultured in fresh medium with or without 5 µM KU55933 for 1 or 2 hr. Values represent the means ± SE from three independent experiments. *p<0.05. The levels of ATM phosphorylation and expression of RAD51 are shown in *Figure 2—figure supplement 2B*. Source data are presented in *Figure 4—source data 1*. (C) ChIP analysis of the RAD51 loading onto the MLL BCR in S412A ARP8 mutant expressing 11–4 Flp-In cells. The experiment was performed as described in (B). Values represent the means ± SE from three independent experiments. n.s: no significant difference. The levels of ATM phosphorylation and RAD51 expression are shown in *Figure 2—figure supplement 2C*. Source data are presented in *Figure 4—source data 1*. (D) ChIP analysis of the RAD51 loading onto the MLL BCR in endogenous ARP8-depleted BIVA cells transfected with either the siRNA-resistant wild-type (WT) or phospho-mimetic ARP8 (S412D). GAPDH is shown as the control region. The control cells were transfected with the empty vector and the non-targeting siRNA (vet). Values represent the means ± SE from three independent experiments. *p<0.05. n.a: not analyzed. Source data are presented in *Figure 4—source data 1*.

DOI: https://doi.org/10.7554/eLife.32222.017

The following source data and figure supplement are available for figure 4:

**Source data 1.** Source raw data for *Figure 4A-D*.
DOI: https://doi.org/10.7554/eLife.32222.019
**Source data 2.** Source raw data for *Figure 4—figure supplement 1B and C*.
DOI: https://doi.org/10.7554/eLife.32222.020
**Figure supplement 1.** Requirement of INO80 and ARP8 for RAD51 binding to the BCR of the MLL gene.
DOI: https://doi.org/10.7554/eLife.32222.018

we examined the involvement of INO80 in RAD51 binding to damaged chromatin. A ChIP assay revealed that the depletion of INO80 by siRNA reduced the loading of RAD51 onto the MLL BCR in BIVA cells after etoposide treatment, suggesting that human INO80 promotes RAD51 binding to the BCR (*Figure 4—figure supplement 1A and B*). We next investigated the requirement for ARP8 in RAD51 binding to the MLL BCR. A ChIP analysis showed that the depletion of ARP8 reduced the binding of RAD51 to the MLL BCR in BIVA cells after etoposide treatment (*Figure 4—figure supplement 1C*). Since the depletion of ARP8 did not affect the level of RAD51 (*Figure 2—figure supplement 1C*), these findings suggest the involvement of ARP8 in the excessive binding of RAD51 to the BCR in BIVA cells.

We then studied the effect of ARP8 phosphorylation on the regulation of RAD51 binding to the BCR after etoposide treatment. The expression of the ARP8 phosphorylation-deficient mutant S412A increased the RAD51 binding to the BCR of the MLL gene after etoposide treatment in ATM-proficient cells (*Figure 4A*). In ATM-proficient cells, ATMi treatment resulted in an increase of RAD51 binding to the BCR (*Figure 4B*), but it did not cause a further increase of RAD51 when ARP8 S412A was expressed (*Figure 4C*). In contrast, the expression of the ARP8 phosphomimetic mutant S412D in ATM-deficient BIVA cells repressed the excessive binding of RAD51 to the BCR (*Figure 4D*). These results strongly suggest that ARP8 phosphorylation at S412 represses the loading of RAD51 onto the MLL BCR after etoposide treatment. Taken together, the phosphorylation of ARP8 regulated by ATM may negatively regulate the loading of RAD51 onto the BCR after etoposide treatment, by repressing the loading of the INO80 complex.

## Repression of 11q23 chromosome translocations through the phosphorylation of ARP8

Having established that ARP8 regulates the loading of RAD51 and INO80 onto the BCR after etoposide treatment, we decided to examine the involvement of ARP8 in 11q23 chromosome translocations. A two-color fluorescence in situ hybridization (FISH) analysis, covering the upstream and downstream regions of the MLL BCR, revealed that the number of BIVA cells carrying split FISH signals after etoposide treatment was significantly reduced by the siRNA-mediated depletion of ARP8, suggesting the involvement of ARP8 in 11q23 chromosomal abnormalities in ATM-deficient cells (*Figure 5A and B*). This finding was confirmed by the FISH analysis with a different DNA probe set (*Figure 5—figure supplement 1A*). In contrast, the number of split signal positive cells among the ATM-proficient 11–4 cells was increased by the depletion of ARP8 (*Figure 5B*). This was confirmed by the FISH analysis of ARP8-deficient Nalm-6 cells (*Figure 5—figure supplement 1B*). These findings suggest that ATM prevents the etoposide-induced 11q23 chromosome translocations through the regulation of ARP8.

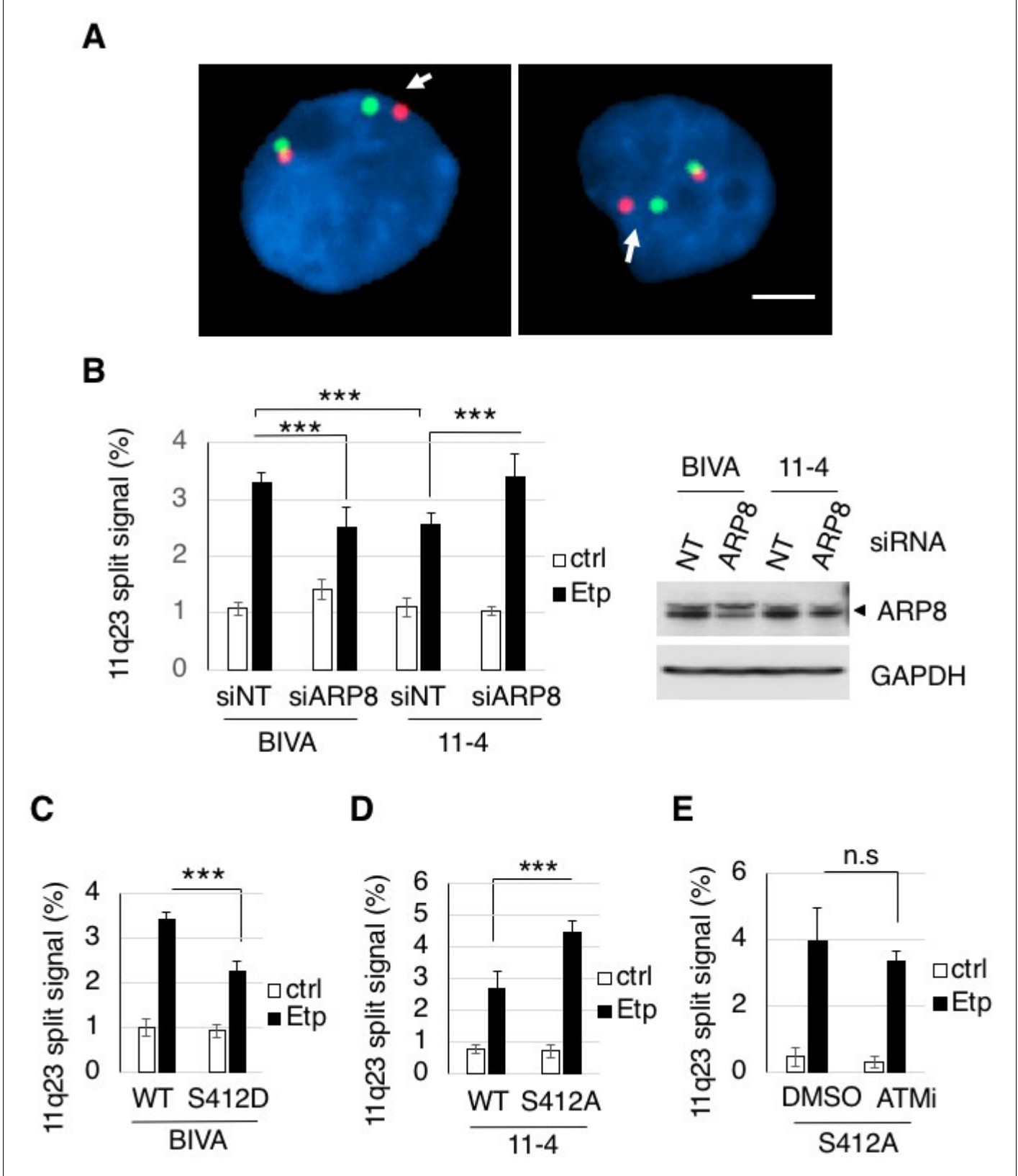

**Figure 5.** ARP8 phosphorylation averts 11q23 chromosome translocations. (**A**) Dual-color FISH analysis of chromosome 11q23. Representative FISH images using etoposide treated BIVA cells are shown. Arrows indicate the split signals (separated by >1 μm). Scale bar: 5 μm. (**B**) The percentages of

*Figure 5 continued on next page*

*Figure 5 continued*

AT5BIVA or 11–4 cells with split chromosome 11q23 signals are shown. The non-targeting control siRNA (siNT) or siARP8-depleted cells were treated with DMSO (ctrl) or etoposide for 15 min, washed, and cultured for 6 hr in fresh medium. At least 2,000 cells were analyzed in every experiment. The average percentages of cells with split signals from four independent experiments are shown. Values represent the means ± SE. ***p<0.001 as determined by the Z test. The ARP8 knockdown is shown in the gel image on the right. Source data are presented in *Figure 5—source data 1*. (C, D) Dual-color FISH analyses of chromosome 11q23 using ARP8-depleted AT5BIVA (C) and 11–4 (D) cells expressing the siARP8-resistant ARP8 wild-type (WT), S412D, or S412A. The average percentages of the cells with split signals from three independent experiments are shown. Values represent the means ± SE. ***p<0.001 as determined by the Z test. Source data are presented in *Figure 5—source data 1*. (E) Dual-color FISH analyses of chromosome 11q23 using 11–4 cells expressing the siARP8-resistant ARP8 S412A. The cells were treated with/without 10 µM ATM inhibitor (KU55933) for 2 hr before etoposide treatment. After the cells were washed, KU55933 (5 µM) was added until the cells were harvested. The average percentages of the cells with split signals from three independent experiments are shown. Values represent the means ± SE from three independent experiments. n. s.: no significant difference. Source data are presented in *Figure 5—source data 1*.

DOI: https://doi.org/10.7554/eLife.32222.021

The following source data and figure supplement are available for figure 5:

**Source data 1.** Source raw data for *Figure 5B-E*.
DOI: https://doi.org/10.7554/eLife.32222.023
**Source data 2.** Source raw data for *Figure 5—figure supplement 1A and B*.
DOI: https://doi.org/10.7554/eLife.32222.024
**Figure supplement 1.** Dual-color FISH analysis of chromosome 11q23 using a different DNA probe set.
DOI: https://doi.org/10.7554/eLife.32222.022

Next, to investigate the role of the ATM-dependent phosphorylation of ARP8 in preventing the 11q23 chromosome abnormalities, we performed the FISH analysis of ATM-deficient BIVA cells expressing the phospho-mimicking ARP8 S412D mutant. The FISH analysis revealed that the expression of ARP8 S412D reduced the number of cells exhibiting split FISH signals after etoposide treatment in ATM-deficient cells (*Figure 5C*). In contrast, the expression of the phospho-deficient ARP8 S412A mutant increased the incidence of 11q23 chromosome translocations in ATM-proficient 11–4 cells (*Figure 5D*). Notably, ATMi treatment failed to enhance the event in ARP8 S412A mutant expressing cells (*Figure 5E*). Taken together, these findings strongly suggest that the ATM-dependent phosphorylation of ARP8 is required to prevent the etoposide-induced 11q23 chromosome abnormalities, through the negative regulation of RAD51 and INO80 binding to the BCR.

## ATM, but not ATR, negatively regulates RAD51 loading onto the BCR after etoposide treatment to repress 11q23 chromosome translocations

Since the etoposide-induced phosphorylation of ARP8 was strongly repressed by an ATR inhibitor in 11–4 cells (*Figure 1E*), the ATR-dependent phosphorylation of ARP8 could also be involved in the excessive RAD51 loading onto the BCR. To test this possibility, we performed a chromatin immunoprecipitation assay of 11–4 cells treated with the ATR inhibitor, VE821. In contrast to the significant increase of etoposide-induced RAD51 binding to the BCR by the treatment with the ATM inhibitor, ATR inhibition failed to so (*Figure 6A*). These results suggest that ATR is not involved in the regulation of RAD51 binding at the BCR of MLL after etoposide treatment.

Next, we examined the effect of an ATRi on the etoposide-induced 11q23 chromosome translocations in ATM-proficient 11–4 cells, by the dual color FISH analysis using the MLL gene probes (*Figure 6B*). The increase of the split signal positive cells by ATRi was less than that by ATMi. Although ATR is suggested to be the major kinase responsible for ARP8 phosphorylation after etoposide treatment, this finding suggests that the effect of ATRi on the etoposide-induced chromosome translocations is limited. Moreover, no additional effects of ATRi on the increase of the chromosome translocations by ATMi were observed. Taken together, these findings strongly suggest that ATM, but not ATR, negatively regulates RAD51 loading onto the BCR after etoposide treatment to repress11q23 chromosome translocations.

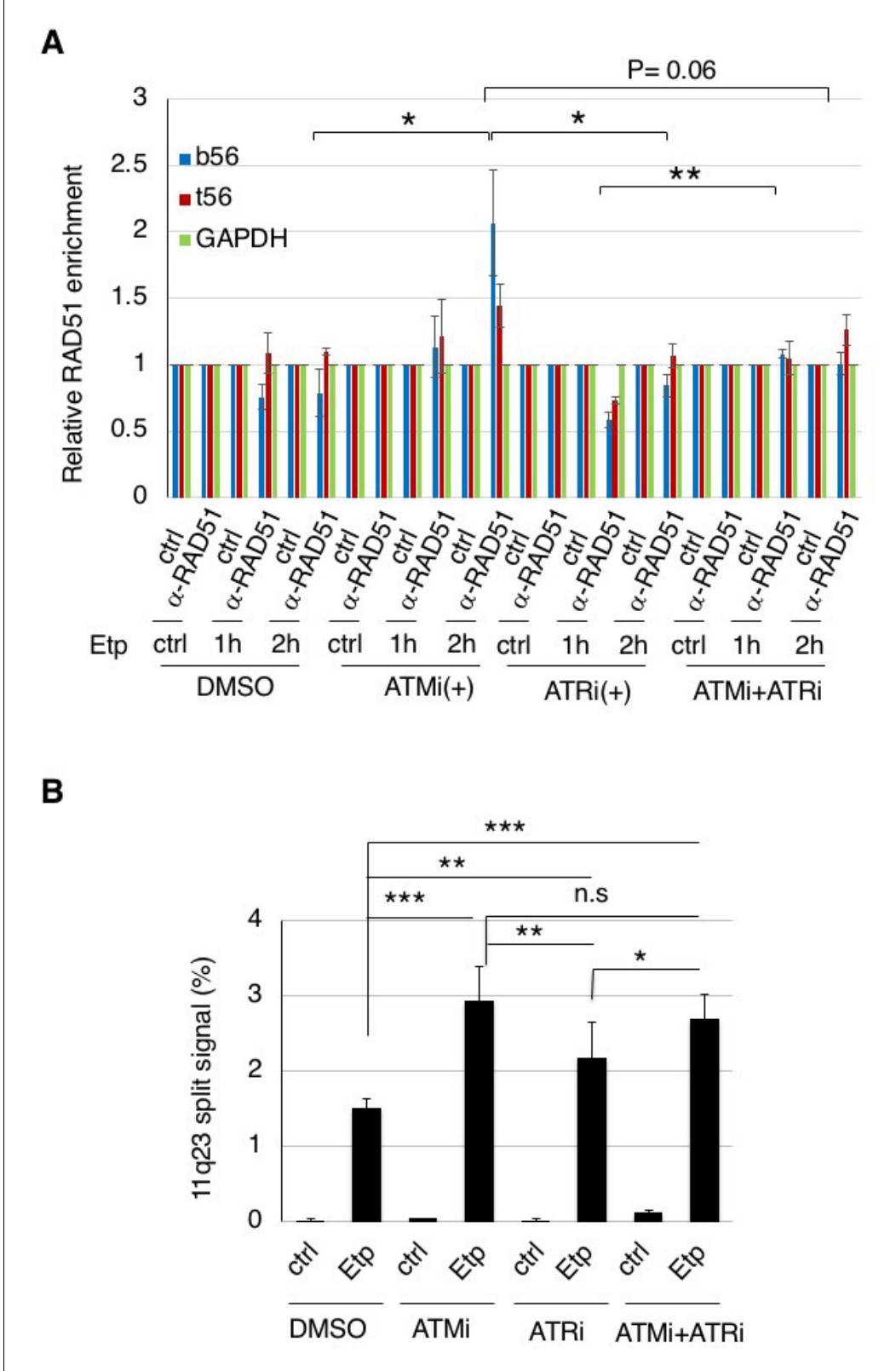

**Figure 6.** ATM, but not ATR, negatively regulates RAD51 loading onto the BCR after etoposide treatment to repress 11q23 chromosome translocations. (**A**) ChIP analysis of the RAD51 loading onto the MLL BCR in ATMi or ATRi or a combination of ATMi and ATRi treated 11–4 cells. 11–4 cells were treated with ATMi (10 µM), ATRi (10 µM), or a combination of ATMi and ATRi for 2 hr before etoposide treatment. After washing the cells, the inhibitors (5 µM) were added until the cells were harvested. The ChIP analysis was performed as described in *Figure 4*. Values represent the

*Figure 6 continued on next page*

*Figure 6 continued*

means ± SE from three independent experiments. *p<0.05. **p<0.01. Source data are presented in *Figure 6—source data 1*. (**B**) The percentages of 11–4 cells with split chromosome 11q23 signals are shown. 11–4 cells were treated with ATMi (10 μM), ATRi (10 μM), or a combination of ATMi and ATRi for 2 hr before etoposide treatment. After the cells were washed, the inhibitors (5 μM) were added until the cells were harvested. Dual-color FISH analyses of chromosome 11q23 were performed as described in *Figure 5*. Values represent the means ± SE from three independent experiments. *p<0.05. **p<0.01, ***p<0.001, n.s.: no significant difference.

DOI: https://doi.org/10.7554/eLife.32222.025

The following source data is available for figure 6:

**Source data 1.** Source raw data for *Figure 6A and B*.

DOI: https://doi.org/10.7554/eLife.32222.026

## Discussion

Our present results are the first to identify ARP8 as the phosphorylation target regulated by ATM and ATR after the induction of DNA damage by etoposide treatment. ARP8 facilitates the binding of INO80 and RAD51 to the BCR of the MLL gene, while the phosphorylation of ARP8 suppresses it through a reduction of its interaction with INO80. The incidence of etoposide-induced 11q23 translocations is reduced by the expression of the phospho-mimicking ARP8 mutant in ATM-deficient cells. Moreover, the expression of the phospho-deficient ARP8 in ATM-proficient cells increases chromosome translocations. ATR was not involved in the regulation of RAD51 binding to the BCR after etoposide treatment. These findings strongly suggest that ATM represses the 11q23 chromosome translocations by regulating the binding of INO80 and RAD51 to the BCR of the MLL gene at functionally appropriate levels, via the phosphorylation of ARP8.

In response to DNA damage, the DNA repair process is facilitated by the phosphorylation of various proteins, including DNA repair factors, cell cycle regulators, and chromatin remodeling factors, through the positive regulation by ATM and ATR (*Maréchal and Zou, 2013*) (*Cimprich and Cortez, 2008*) (*Clouaire et al., 2017*; *Shiloh, 2003*; *Shiloh and Ziv, 2013*). In contrast, Exo1, an exonuclease for end resection in HR, is also phosphorylated by ATM in response to DNA damage, but to inhibit its exonuclease activity for the prevention of the untimely generation of ssDNA for RAD51 loading (*Bolderson et al., 2010*). In this study, we show that the phosphorylation of ARP8 represses the binding of INO80 and RAD51 to damaged chromatin (*Figures 2* and *4*). Importantly, the phosphorylation of ARP8 reduced its interaction with INO80 (*Figure 3*, *Figure 3—figure supplements 1* and *2*). Since ARP8 is required for the binding of the INO80 complex to damaged chromatin (*Kashiwaba et al., 2010*; *Saravanan et al., 2012*), the reduced interaction of ARP8 with INO80 by its phosphorylation may repress the binding of INO80 to the damaged chromatin and thus reduce ssDNA formation and RAD51 loading around damaged sites. ARP8 phosphorylation is significant from 2 hr after etoposide treatment, which is slower than the phosphorylation of ATM and H2AX. Therefore, it may also facilitate the dissociation of INO80 from damaged chromatin after the appropriate remodeling of damaged chromatin for DNA repair to avoid illegitimate recombination, leading to chromosome abnormalities. The phosphorylation of various proteins regulated by ATM may play an important role to prevent chromosome abnormalities, by maintaining the recombination activity within an appropriate range through both positive and negative regulation of repair proteins.

The phosphorylation of ARP8 is relatively slow, and occurs more than 2 hr after etoposide treatment, as compared to the early activation of ATM within 1 hr (*Bakkenist and Kastan, 2003*; *Tanaka et al., 2007*). Since the recombinational repair of DSBs starts slowly as compared to the end-joining repair, which normally begins within 30 min (*Mao et al., 2008*), this is consistent with the notion that ARP8 plays an important role in the appropriate regulation of the recombinational repair proteins. The slower phosphorylation of ARP8 also suggests the involvement of kinases other than ATM. Indeed, we found that ATR also regulates the etoposide-induced phosphorylation of ARP8. Moreover, ATRi repressed the phosphorylation of ARP8 especially from 2 hr after the etoposide treatment. These findings suggest the presence of different regulation systems of ARP8 phosphorylation after the induction of DNA damage. The phosphorylation of ARP8 after etoposide treatment

could be regulated by multiple steps and factors for the precise control of DNA repair activity to maintain chromosome stability.

This study revealed that ATR is likely to be the major kinase responsible for the etoposide-induced ARP8 phosphorylation. However, unlike the inhibition of ATM, ATR inhibition failed to increase the RAD51 binding to the BCR of MLL after etoposide treatment (*Figure 6*), suggesting that ATR is not involved in the regulation of RAD51 binding to the BCR after etoposide treatment. Although the mechanism of 11q23 chromosome translocations is still unclear, a specific DNA sequence and/or chromatin structure of the BCR has been suggested to promote the mis-rearrangement of this region during the repair process (*Gole and Wiesmüller, 2015*). Therefore, ARP8 phosphorylation by ATM functions in the prevention of chromosome translocation at the BCR, while that by ATR may play roles in the repair of different types of DNA damage not relevant to chromosome translocations. Further studies are required to clarify the distinct roles of ATM and ATR in the phosphorylation of ARP8 after the induction of DNA damage, to coordinate the HR activity for accurate DNA repair.

Several lines of evidence have suggested the association of HR with genomic instability (*Bishop and Schiestl, 2003*; *Guirouilh-Barbat et al., 2014*; *Mizuno et al., 2009*; *Reliene et al., 2007*; *Ruiz et al., 2009*). RAD51-deficient vertebrate cells accumulate chromosomal breaks, resulting in an early embryonic lethal phenotype (*Lim and Hasty, 1996*; *Sonoda et al., 1998*). However, the overexpression of human RAD51 also leads to genomic instability (*Kim et al., 2001*; *Marsden et al., 2016*; *Reliene et al., 2007*; *Richardson et al., 2004*). These findings suggest that both the down- and up-regulation of the recombination activities through the levels of the RAD51 protein are associated with chromosomal instability. Together with the regulation of the RAD51 function at the protein level, the regulation of the BCR binding by the ATM-regulated phosphorylation of ARP8 may play an important role in maintaining the local recombination activities within an appropriate range, to ensure the fidelity of DNA repair and prevent chromosome translocations. INO80 and ARP8 have been shown to regulate the RAD51 loading to damaged chromatin in yeast (*Tsukuda et al., 2005*) (*Tsukuda et al., 2009*) (*van Attikum et al., 2007*) (*Lademann et al., 2017*). Moreover, the overexpression of human RAD51 leads to the various types of chromosome abnormalities (*Reliene et al., 2007*; *Richardson et al., 2004*). Therefore, this regulation of HR by ARP8 in human cells may also be applicable to DSBs in general.

The wild-type and the phospho-deficient mutant of ARP8 show increased interactions with INO80 after etoposide treatment, but the phospho-mimetic mutant does not (*Figure 3*, and *Figure 3—figure supplements 1* and *2*). Human ARP8 contains five insertions in the conserved actin fold domain (*Gerhold et al., 2012*) (*Figure 1—figure supplement 1B*). The S412 residue is located in a major loop insertion (insertion IV, residues 401–507) (*Figure 1—figure supplement 1C*). This region lacks interpretable electron density, suggesting its high flexibility to mediate dynamic protein–protein interactions (*Gerhold et al., 2012*). Moreover, the involvement of insertion IV in forming a proper ARP8-DNA complex has been suggested (*Osakabe et al., 2014*). Therefore, the phosphorylation of S412 within insertion IV of ARP8 could affect the binding activity of the INO80 complex to damaged chromatin, by suppressing the interactions with INO80 and components of damaged chromatin. Interestingly, ARP8 is conserved from yeast to human, but the 412-SQ motif is conserved only in higher eukaryotes, and not in yeast, Drosophila and Xenopus (*Figure 1—figure supplement 1D*). Although the mechanism by which the ARP8 phosphorylation regulates the activity of the INO80 complex remains to be clarified, the DNA damage-dependent ARP8 phosphorylation may have evolutionary advantages in DNA repair.

Chromosome abnormalities involving the MLL gene are one of the most frequent chromosomal aberrations observed in secondary leukemia associated with cancer therapy. We have shown that the etoposide-induced DNA damage in ATM-deficient cells facilitates the illegitimate recombination at the MLL gene, through the excessive binding of RAD51 and INO80 to the BCR. Our findings highlight the importance of the ATM-dependent modulation of recombination repair to avert 11q23 chromosome translocations. Further studies to investigate the mechanism that maintains the fidelity of DNA repair activity, using the chromosome translocations observed in secondary malignancy will provide new insights into the general mechanisms of carcinogenesis.

# Materials and methods

## Key resources table

| Reagent type (species) or resource | Designation | Source or reference | Identifiers | Additional information |
|---|---|---|---|---|
| Gene (*Homo Sapiens*) | arp8 | NA | GenBank: GeneID 93973 | |
| Gene (*Homo Sapiens*) | ino80 | NA | GenBank: GeneID 54617 | |
| Gene (*Homo Sapiens*) | atm | NA | GenBank: GeneID 472 | |
| Gene (*Homo Sapiens*) | atr | NA | GenBank: GeneID 545 | |
| Gene (*Homo Sapiens*) | rad51 | NA | GenBank: GeneID 5888 | |
| Gene (*Homo Sapiens*) | rpa2 | NA | GenBank: GeneID 6118 | |
| Cell line (*Homo Sapiens*) | AT5BIVA | PMID:21048951 | RRID: CVCL_7442 | Cell line maintained in S. Matsuura lab; |
| Cell line (*Homo Sapiens*) | 11–4 | PMID:21048951 | | Cell line maintained in S. Matsuura lab; |
| Cell line (*Homo Sapiens*) | U2OS | ATCC | RRID: CVCL_0042 | |
| Cell line (*Homo Sapiens*) | GM0637 | other | RRID: CVCL_7297 | Cell line maintained in T. Cremer lab; |
| Cell line (*Homo Sapiens*) | Tet-Off ARP8 Nalm-6 | PMID:25299602 | | Cell line maintained in M. Harata lab; |
| Transfected construct (*Homo Sapiens*) | Flp-In T-REx 11–4 | this paper | | Progenitor = 11–4; constructed by use of Flp-In T-REx core kit (Invitrogen) |
| Transfected construct (*Homo Sapiens*) | Flp-In T-REx 11–4 HA-ARP8-WT | this paper | | Progenitor = Flp In T-REx 11–4; Addition of tetracyclin induces expression of HA-tagged recombinant ARP8 wild-type protein |
| Transfected construct (*Homo Sapiens*) | Flp-In T-REx 11–4 HA-ARP8-S412A | this paper | | Progenitor = Flp In T-REx 11–4; Addition of tetracyclin induces expression of HA-tagged recombinant ARP8-S412A mutant protein |
| Transfected construct (*Homo Sapiens*) | U2OS HA-ARP8-WT | this paper | | Progenitor = U2 OS; stably expressing HA-tagged recombinant ARP8 wild-type protein |
| Transfected construct (*Homo Sapiens*) | U2OS HA-ARP8-S412A | this paper | | Progenitor = U2 OS; stably expressing HA-tagged recombinant ARP8-S412A protein |
| Antibody | anti-phospho ATM/ATR substrate motif (rabbit monoclonal) | Cell Signaling Technology | Cell Signaling Technology :Cat# 6966S; RRID:AB_10949894 | |
| Antibody | anti-ATM protein kinase pS1981 (mouse monoclonal) | Rockland | Rockland:Cat# 200-301-400; RRID:AB_217868 | |
| Antibody | anti-HA Tag (mouse monoclonal) | Merck Millipore | Merck Millipore: Cat# 05–904; RRID:AB_417380 | |
| Antibody | anti-Histone H2A.X, phospho (Ser139) (mouse monoclonal) | Merck Millipore | Merck Millipore: Cat# 05–636; RRID:AB_309864 | |
| Antibody | anti-human RAD51 (rabbit polyclonal) | Bio Academia | Bio Acdemia: Cat# 70–001; RRID:AB_2177110 | |

*Continued on next page*

*Continued*

| Reagent type (species) or resource | Designation | Source or reference | Identifiers | Additional information |
|---|---|---|---|---|
| Antibody | anti-phospho RPA32 (S4/S8) (rabbit polyclonal) | Bethyl Laboratories | Bethyl Laboratories: Cat# A300-245A; RRID:AB_210547 | |
| Antibody | anti-INO80 (rabbit polyclonal) | Bethyl Laboratories | Bethyl Laboratories: Cat# A303-371A; RRID:AB_10950580 | |
| Antibody | anti-Human RPA/p34 (Replication Protein A) Ab-1 (mouse monoclonal) | Lab Vision | Lab Vision: Cat# MS-691-P0; RRID:AB_143149 | |
| Antibody | anti-GAPDH (mouse monolconal) | Santa Cruz Biotechnology | Santa Cruz Biotechnology: sc-32233; RRID:AB_627679 | |
| antibody | anti-Histone H2A.X | Abcam | Abcam: Cat# ab124781; RRID_AB_10971675 | |
| antibody | anti-beta-actin (mouse monoclonal) | Sigma-Aldrich | Sigma-Aldrich:Cat # A5441; RRID:AB_476744 | |
| antibody | anti-ARP8 (rabbit monoclonal) | PMID:25299602 | | |
| antibody | Cy3-secondary | Invitrogen | | |
| Recombinant DNA reagent | pME18FL-hARP8 | PMID:18163988 | | |
| Recombinant DNA reagent | pcDNA3.1/Myc-His(-) | Invitrogen | Invitrogen:Cat # V855-20 | |
| Recombinant DNA reagent | pcDNA5/FRT/TO | Invitrogen | Invitrogen:Cat # K650001 | |
| Recombinant DNA reagent | HA-arp8 | this paper | | Progenitor = pME18FL-hARP8; PCR, HA tag was fused; mutagenized in the Ambion Silencer Select s41201 siRNA target site for resistance; inserted into pcDNA3.1/Myc-His(-) or pcDNA5/FRT/TO |
| Recombinant DNA reagent | HA-arp8-S412A | this paper | | Progenitor = HA-arp8; PCR, mutagenized; inserted into pcDNA3.1/Myc-His(-) or pcDNA5/FRT/TO |
| Recombinant DNA reagent | HA-arp8-S412D | this paper | | Progenitor = HA-arp8; PCR, mutagenized; inserted into pcDNA3.1/Myc-His(-) |
| Sequence-based reagent | oligonucleotide for construction of siRNA-resistant HA-ARP8 | this paper | | 5'-CTCAACAAAATGCCACCATCG TTCAGACGTATAATTGAAAATG TGGATG-3' |
| Sequence-based reagent | oligonucleotide for construction of siRNA-resistant HA-ARP8-S412A | this paper | | 5'-TTGCAGCACAGAGCTCAGGGCG ATCCTG-3' |
| Sequence-based reagent | oligonucleotide for construction of siRNA-resistant HA-ARP8-S412D | this paper | | 5'-TTGCAGCACAGAGATCAGG GCGATCCTG-3' |
| Sequence-based reagent | primer for RT-PCR of bt56 forward | PMID:21048951 | | 5'-TACTCTGAATCTCCCGCA-3' |
| Sequence-based reagent | primer for RT-PCR of bt56 reverse | PMID:21048951 | | 5'-CGCTCGTTCTCCTCTAA-3' |

*Continued on next page*

*Continued*

| Reagent type (species) or resource | Designation | Source or reference | Identifiers | Additional information |
|---|---|---|---|---|
| Sequence-based reagent | primer for RT-PCR of t56 forward | PMID:21048951 | | 5'-TTGCCAAGTCTGTTGTGAG-3' |
| Sequence-based reagent | primer for RT-PCR of t56 reverse | PMID:21048951 | | 5'-CAGAGGCCCAGCTGTAGTTC-3' |
| Sequence-based reagent | primer for RT-PCR of GAPDH forward | this paper | | 5'-TCTCCCCACACACATGCACTT-3' |
| Sequence-based reagent | primer for RT-PCR of GAPDH reverse | this paper | | 5'-CCTAGTCCCAGGGCTTTGATT-3' |
| Sequence-based reagent | primer for RT-PCR of beta-globin forward | PMID:21048951 | | 5'-TTGGACCCAGAGGTTCTTTG-3' |
| Sequence-based reagent | primer for RT-PCR of beta-globin reverse | PMID:21048951 | | 5'-GAGCCAGGCCATCACTAAAG-3' |
| Commercial assay or kit | Duolink PLA | Sigma-Aldrich | Sigma-Aldrich:Cat # DUO92002, DUO92004, DUO92008 | Proximity Ligation Assay |
| Chemical compound, drug | etoposide | Sigma-Aldrich | Sigma-Aldrich:Cat # E1383 | |
| Chemical compound, drug | ATM inhibitor (KU55933) | Merck Millipore | Merck Millipore:Cat# 118500 | |
| Chemical compound, drug | ATR inhibitor IV | Merck Millipore | Merck Millipore:Cat# 504972 | |
| Software, algorithm | Metefer 4 MetaCyte | Metasystems | v 3.11.4 | software for FISH analysis |
| Software, algorithm | Image J | NIH | | |
| Other | XL MLL Plus | Metasystems Probes | Metasystems Probes: Cat# D5060-100-OG | probe for FISH analysis |
| Other | LSI MLL Dual Color, BreakApart Rearrangement Probe | Vysis, Abbott Molecular Inc. | Vysis, Abbott Molecular Inc.: 32–190083 | probe for FISH analysis |

## Cell culture and chemical treatment

The SV40-transformed AT fibroblast cell line AT5BIVA and its ATM-proficient derivative, AT5BIVA cells reconstituted with chromosome 11 (11-4), were kindly provided by Dr. S. Matsuura (*Sun et al., 2010*). The AT5BIVA cells were cultured in Dulbecco's Modified Eagle's medium (DMEM, Sigma-Aldrich), supplemented with 10% fetal bovine serum (FBS, Equitech-Bio, Kerrville, USA). The 11–4 cells were maintained in DMEM supplemented with 10% FBS and 0.2 mg/ml of G418 (Nacalai Tesque). The Flp-In T-Rex 11–4 cells were maintained in DMEM, supplemented with 10% tetracycline-free FBS (Sigma-Aldrich, St. Louis, USA), 10 µg/ml blasticidin S HCl (Gibco, Japan) and 40 µg/ml hygromycin B (Invitrogen, Carlsbad, USA). The human osteosarcoma U2OS cell line (ATCC) was cultured in Minimum Essential Medium Eagle (MEM, Sigma-Aldrich), supplemented with 10% FBS. GM0637 cells were cultured in DMEM, supplemented with 10% FBS. Tet-Off ARP8 Nalm-6 cells were cultured at 37°C in RPMI-1640, containing GlutaMAX-I (Gibco) and supplemented with 10% FBS. For the induction of the ARP8 knockout, tetracycline (Sigma-Aldrich) was added to the culture medium to a final concentration of 2 µg/ml (*Osakabe et al., 2014*). AT5BIVA, 11–4 cell lines were kindly provided by Dr. S. Matsuura laboratory, Hiroshima University, Japan. U2OS cell line was purchased from ATCC. GM0637 cell line was kindly provided by Dr. T. Cremer laboratory, LMU, Germany. Tet-Off ARP8 Nalm-6 cell line was kindly provided by Dr. M. Harata laboratory, Tohoku University, Japan. For the induction of DNA damage, as described elsewhere, the cells were exposed to 100 µM etoposide (Sigma-Aldrich), unless otherwise stated, for 15 min, washed, and cultured in fresh medium. Dimethylsulfoxide (DMSO) was used as the vehicle for etoposide, and was present in the cell cultures at a final concentration of 0.1%. Unless otherwise stated, both the ATM inhibitor KU55933 (Merck Millipore, Billerica, USA) and ATR inhibitor IV VE821 (Merck Millipore) were used at 10 µM for two hours before etoposide treatment and at 5 µM after the cells were washed, respectively.

## Antibodies

The antibodies used for chromatin immunoprecipitation, immunoblotting, and immunofluorescence staining were rabbit anti-phospho ATM/ATR substrate motif (Cell Signaling Technology, Danvers, USA), mouse anti-phospho ATM (Rockland, Pottstown, USA), mouse anti-HA, mouse anti-γH2AX (Merck Millipore), rabbit anti-RAD51 (Bio Academia,Japan), rabbit anti -RPA2 S4/8 and rabbit anti-INO80 (Bethyl, Montgomery, USA), mouse anti-RPA34 (Lab Vision, Fremont, USA), rabbit anti-glyceraldehyde-3-phosphate dehydrogenase (anti-GAPDH) (Santa Cruz Biotechnology, USA), rabbit anti-H2AX (Abcam, UK), and mouse anti-β-actin (Sigma-Aldrich) and rabbit anti-ARP8 (*Osakabe et al., 2014*).

## RNAi and plasmids

All siRNAs were Ambion Silencer Select siRNAs (Thermo Fisher Scientific,USA). The siRNAs were s41201 for ARP8, s57219 for ATM, s3638 for CK2, s29257 and s224310 for INO80. Select Negative Control siRNA was used as the control. The siRNA interference experiments were performed 2 days after transfection with 0.2–0.3 nM siRNA, using Lipofectamine RNAiMAX (Invitrogen). The pcDNA 3.1 vector bearing the HA-tagged ARP8 was constructed by inserting the PCR-amplified ARP8 cDNA into the *Not*I/*Hind*III sites, followed by inserting the PCR-amplified HA between the *Apa*I and *Xho*I sites of pcDNA3.1/Myc-His (-) C. Plasmid transfections were performed using the GeneJuice transfection reagent (Novagen, Billerica, USA), according to the manufacturer's instructions. For rescue experiments, the siRNA-resistant ARP8 expression vector was co-transfected with the siRNA, using the Lipofectamine 2000 transfection reagent (Invitrogen) for 2 or 3 days. The mutant vectors were constructed by site-directed mutagenesis, using the indicated oligonucleotides: For the siRNA-resistant HA-ARP8 mutant, 5'-ctcaacaaaatgccaccatcgttcagacgtataattgaaaatgtggatg-3', for the HA-ARP8 S418A mutant, 5'-TTGCAGCACAGAGCTCAGGGCGATCCTG-3', and for the S418D mutant, 5'-TTGCAGCACAGAGATCAGGGCGATCCTG-3'. The sequences were confirmed using a BigDye Terminator v3.1 Cycle Sequencing Kit and an Applied Biosystems Genetic Analyzer, model 3130.

## Establishment of stably expressing cells

To generate cells stably expressing U2OS, the pcDNA 3.1 plasmid encoding the HA-tagged wild-type or S412A mutant of ARP8 was transfected using the GeneJuice transfection reagent, with G418 selection (Nacalai Tesque, Japan). After the confirmation of stable expression by immunofluorescence staining and immunoblotting, pools of single clones were used for experiments. For the generation of the inducible expression of wild-type or phosphorylation mutant ARP8 in 11–4 cells, the Flp-In System (Thermo Fisher Scientific) was used, according to the manufacturer's instructions. The siRNA-resistant HA-tagged wild-type,or S412A ARP8 fragment from pcDNA 3.1 HA-ARP8 was inserted between the *Hind*III and *Kpn*I sites of the pcDNA5/FRT/TO vector. For the induction of the HA-ARP8 expression, a final concentration of 2 μg/ml tetracycline was added for 24 hr.

## FISH analysis

FISH analyses were performed using the 11q23 chromosome translocation probe (XL MLL plus, MetaSystems probes, Germany) and the LSI MLL Dual Color, BreakApart Rearrangement Probe (Vysis, Abbott Molecular Inc. Abbott park, USA), according to the manufacturers' protocols. For the DNA FISH analysis using the probe from Metaystems, the images were acquired on an Axio Imager Z2 microscope (Carl Zeiss, Germany) equipped with the MetaSystems software. Subsequently, at least 2000 etoposide-exposed or DMSO-exposed cells were counted by the Metafer platform, Meta-Cyte, and the cells containing split signals (separated by >1 μm) were monitored. For the DNA FISH analysis using the probe from Vysis, the images were acquired on a Zeiss AxioplanII microscope using an AxioCamMRm controlled by Axiovision. At least 200 etoposide-exposed or DMSO-exposed cells were counted. All FISH analyses were repeated three times.

## Chromatin immunoprecipitation and real-time PCR assay

In brief, the cells were fixed by adding formaldehyde to a 1% final concentration for 10 min at 25℃. The cells were then sonicated to prepare chromatin suspensions of DNA fragments that were roughly 300–500 bps in length. Immunoprecipitations were performed using antibodies against INO80 and RAD51. Normal rabbit IgG was used as the negative control. Real-time PCR reactions

were performed using SYBR premix Ex Taq (TAKARA, Japan). The dissociation curve analysis of the melting temperature of the amplified DNA showed that each primer set gave a single, specific product. The immunoprecipitation data were normalized to those of a control region in the GAPDH or β-globin gene, to correct for experimental variation. The relative immunoprecipitation value represents the ratio of the immunoprecipitated DNA after chemical treatment to the immunoprecipitated DNA after vehicle treatment. All ChIP analyses were repeated at least three times, and in each experiment, quantitative PCR reactions were performed in duplicate. Values represent the means ± SE. The primers for real-time PCR were: bt56 forward: 5 '-TACTCTGAATCTCCCGCA-3' bt56 reverse: 5'-CGCTCGTTCTCCTCTAA-3' t56 forward: 5'-TTGCCAAGTCTGTTGTGAGC-3' t56 reverse: 5'-CAGAGGCCCAGCTGTAGTTC-3'

GAPDH forward: 5'-TCTCCCCACACACATGCACTT-3'
GAPDH reverse: 5'-CCTAGTCCCAGGGCTTTGATT-3'.
β-globin forward: 5'-TTGGACCCAGAGGTTCTTTG3'
β-globin reverse: 5'-GAGCCAGGCCATCACTAAAG3'

## Immunofluorescence staining

After fixation in 4% paraformaldehyde in 1X phosphate-buffered saline (PBS) for 10 min at room temperature, the cells were permeabilized with 0.1% sodium dodecyl sulfate (SDS)−0.5% Triton X-100 in 1x PBS for 5 min. For the detection of HA-tagged ARP8, fixed cells were incubated for 30 min at 37°C with a mouse anti-HA antibody (1:600) in 1% bovine serum albumin (BSA)/1 XPBS. Cy3-conjugated goat anti-mouse (1:1,000, Invitrogen) antibodies were used as the secondary antibodies. Cells were mounted using Vectashield containing DAPI and observed with a BZ-X700 microscope (Keyence).

## Immunoprecipitation and immunoblotting

The nuclear fraction was prepared as described previously (*Liu et al., 2015*). The whole cells extracts were prepared using buffer C (20 mM HEPES pH 7.9, 400 mM NaCl, 1 mM EDTA, 0.1% NP-40, 0.5 mM DTT, 1X protease inhibitor (Roche), 1X phosphatase inhibitor (Nacalai Tesque), 20% glycerol), and the diluted lysates were used for immunoprecipitation. For immunoprecipitation assay of transfected cells, anti-HA antibody conjugated IgG Dynabeads (Novex) were used, and for the endogenous protein, anti-INO80 antibody, anti-ARP8 antibody, or normal IgG conjugated IgG Dynabeads were used. The immunoprecipitation analysis was performed at least twice to confirm the results. The precipitates were electrophoresed through a gel and probed by western blotting with the indicated the antibodies. The intensities of the bands were quantified, using the Image J software.

## Proximity ligation assay (PLA)

GM0637 cells were transfected with the HA-tagged wild type or S412A or S412D mutant of ARP8 or empty vector for 24 hr, fixed with 2% paraformaldehyde in PBS for 10 min at room temperature and permeabilized with 0.5% Triton X-100 in PBS for 5 min at room temperature. The cells were incubated with the rabbit anti-INO80 and mouse anti-HA antibodies diluted in 0.1% BSA for 30 min at 37°C in a moist chamber. Proximity ligation was then conducted in situ, according to the manufacturer's instructions (Olink Bioscience, Sweden). We used the PLA probe anti-rabbit PLUS and the PLA probe anti-mouse MINUS. To visualize the interaction between two proteins, the samples were incubated for ligation and amplification. After serial SSC (sodium/sodium citrate) washes, nuclei were stained with DAPI. The slides were mounted with Vectashield (Vector Labs). PLA signals were detected with an LSM780 confocal microscope (Carl Zeiss), with a 63 × 1.40 NA plan-apochromat objective, and counted in at least 400 cells with the Image J software. All PLA analyses were repeated four times.

## Data analysis

Data in all experiments are represented as mean ± SE. Statistical analysis was performed using the two-tailed unpaired t- test. For the FISH analysis, the percentages of cells with split signals were determined by the Z test of homogeneity for independent samples. Results reaching $p < 0.05$ were considered to be statistically significant (*$p < 0.05$, **$p < 0.01$, ***$p < 0.001$).

## Acknowledgements

This research was supported by JSPS KAKENHI Grants, number JP26430114 to J S, and numbers JP16H01312 and JP15H02821 to ST. This work was partially supported by the Program of the network-type joint Usage/Research Center for Radiation Disaster Medical Science of Hiroshima University, Nagasaki University and Fukushima Medical University, and the Program of the Joint Usage/Research Center of Kyoto University supported by the Ministry of Education, Culture, Sports, Science and Technology (MEXT) of Japan. RK is a member of the Oncode Institute which is partly financed by the Dutch Cancer Society and the gravitation program CancerGenomiCs.nlfrom the Netherlands Organisation for Scientific Research (NWO). We thank S Matsuura and H Kawai for cell lines, H Kurumizaka for suggestions, H Shimamoto for data analysis and H Yamada for manuscript proofreading.

## Additional information

### Funding

| Funder | Grant reference number | Author |
| --- | --- | --- |
| Japan Society for the Promotion of Science | JP26430114 | Jiying Sun |
| Ministry of Education, Culture, Sports, Science, and Technology | the Program of the network-type joint Usage/Research Center for Radiation Disaster Medical Science o | Satoshi Tashiro |
| Japan Society for the Promotion of Science | JP16H01312 | Satoshi Tashiro |
| Japan Society for the Promotion of Science | JP15H02821 | Satoshi Tashiro |
| Ministry of Education, Culture, Sports, Science, and Technology | Program of the Joint Usage/Research Center of Kyoto University | Satoshi Tashiro |
| KWF Kankerbestrijding | The Oncode Institute | Roland Kanaar |
| Nederlandse Organisatie voor Wetenschappelijk Onderzoek | The gravitation program CancerGenomiCs.nl | Roland Kanaar |

The funders had no role in study design, data collection and interpretation, or the decision to submit the work for publication.

### Author contributions

Jiying Sun, Conceptualization, Supervision, Writing—original draft, Project administration, Writing—review and editing; Lin Shi, Validation, Methodology, Writing—original draft; Aiko Kinomura, Yukako Oma, Validation, Methodology, Writing—review and editing; Atsuhiko Fukuto, Masahiko Harata, Tsuyoshi Ikura, Methodology; Yasunori Horikoshi, Masae Ikura, Roland Kanaar, Methodology, Writing—review and editing; Satoshi Tashiro, Writing—review and editing

### Author ORCIDs

Jiying Sun (iD) http://orcid.org/0000-0002-0563-1245
Roland Kanaar (iD) https://orcid.org/0000-0001-9364-8727
Satoshi Tashiro (iD) http://orcid.org/0000-0001-7331-157X

### Decision letter and Author response

Decision letter https://doi.org/10.7554/eLife.32222.031
Author response https://doi.org/10.7554/eLife.32222.032

## Additional files

### Supplementary files
• Transparent reporting form
DOI: https://doi.org/10.7554/eLife.32222.027

### Data availability
All data generated or analysed during this study are included in the manuscript and supporting files. Source data files have been provided.

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
