## [Decision Letter]

Thank you for submitting your article "ATM phosphorylates ARP8 to prevent chromosome translocations by counteracting repair protein loading to damaged sites" for consideration by *eLife*. Your article has been reviewed by three peer reviewers, and the evaluation has been overseen by Jerry Workman as the Reviewing Editor and Jessica Tyler as the Senior Editor. The reviewers have opted to remain anonymous.

The reviewers have discussed the reviews with one another and the Reviewing Editor has drafted this decision to help you prepare a revised submission.

Summary:

In this manuscript, the authors addressed the phosphorylation of ARP8, a subunit of INO80-C, upon etoposide treatment, and how this phosphorylation affects the MLL translocation, which occurs frequently in leukemia. The authors established that ATM regulates phosphorylation of ARP8 S412 after etoposide treatment by an anti-ATM/ATR substrate antibody. Arp8 phosphorylation detected by this antibody was sensitive to the phosphatase treatment and it was abolished upon mutation of ARP8 S412 to alanine, suggesting that S412 is the only phosphorylation site detectable in ARP8. They further showed that S412 phosphorylation is sensitive to treatment with the ATM inhibitor KU55933 and to knockdown of ATM siRNA, suggesting that ATM regulates S412 phosphorylation.

To explore the molecular role of ARP8 S412 phosphorylation, the authors first tested its influence on INO80-ARP8 protein interaction upon etoposide treatment. They found that ATM inhibition or ARP8 alanine mutation at S412 of ARP8 enhanced INO80-ARP8 interaction. These findings indicate that ATM dependent phosphorylation on ARP8-S412 reduces ARP8 INO80 interaction.

To elucidate consequences of changed ARP8 INO80 interaction, the authors investigated the binding of INO80 and a HR protein RAD51 to a chromosome breakpoint clustered in BCR MLL locus that is associated with chromosome translocation. They demonstrated by ChIP analyses that increased INO80 and RAD51 association to the break point upon etoposide treatment is sensitive to ARP8 silencing and phospho-mimetic S412D ARP8 mutation in ATM deficient BIVA cells. Conversely, they showed that an ARP8 S412A non-phospho mutation increases INO80 and RAD51 binding to the break point in ATM-proficient cells. These results suggest that the phosphorylation of ARP8 S412, which minimizes the INO80 interaction, reduces INO80 and RAD51 association to the break point.

The authors further tested the ARP8 mutants on translocation. They showed by FISH analysis that 11q23 chromosome translocations after etoposide treatment in ATM-deficient cells (BIVA) are significantly reduced upon siRNA-mediated knockdown of ARP8, whereas translocations are increased in ATM-proficient cells (11-4). Expression of the ARP8 S412A or D mutant affects the number of chromosome translocations in an opposite manner, consistent with the notion that the ARP8 S412 phosphorylation reduces translocation events.

Overall, the data presented in the manuscript support a model that ARP8 phosphorylation mediated by ATM suppresses 11q23 translocation by reducing the binding of the INO80 and RAD51 HR protein to the break site. However, authors did not provide any evidence that ATM directly targets ARP8, nor ATM and ARP8 function in the same pathway.

Essential revisions:

1) In the current manuscript, we did not find data that shows direct phosphorylation of ARP8 S412 by ATM, as the title of this manuscript indicates. The antibody probes for ATM or ATR targets and ATM inhibition only partially inhibits ARP8 phosphorylation detected by the antibody. Furthermore, ARP8 S412 phosphorylation status in ATM deficient BIVA cells compared to the ATM proficient cells was not shown. The onset of S412 phosphorylation after etoposide treatment is rather late. To correlate ATM action in this event, authors should compare the kinetics of ATM activation and another ATM target in parallel to ARP8 S412. As in the Discussion, it is plausible that ATR is also responsible for the ARP8 S412 phosphorylation. Phosphorylation analysis in combination with an ATR inhibitor would help to answer this question. To evaluate the phenotype that is influenced by ARP8 S412 phosphorylation and ATM, you should analyze the epistatic relation. Is S412A mutation additive or not to the ATM deficiency or ATM inhibition? For example, does ARP8 S412A mutation increase INO80 and RAD51 binding to the break point and 11q23 translocation increased in BIVA cells or by ATM inhibition in ATM proficient cells? Basically, we feel that there are many over-interpretations of these data throughout the manuscript (title, subtitles, figure legends etc.).

Under the current data set, the authors should not state "ATM phosphorylates ARP8", rather than "ATM regulates ARP8 phosphorylation", otherwise authors should provide an in vitro ATM kinase assay on ARP8.

2) Data showing that ARP8 phosphorylation is attenuated by ATMi could be bolstered by examining phosphorylation status of ARP8 in ATM-deficient cells (ATVBIVA). A possible role of ATR in this phosphorylation should be investigated at least by using specific ATRi.

3) Since gammaH2AX recruits INO80 to DSBs, the ChIP analyses in Figure 2 could be extended to gammaH2AX to confirm that etoposide induces DSBs specifically at the BCR.

4) Does the S412D mutant fail to interact with INO80 in co-IPs? That could help explain the lack of INO80 enrichment in Figure 2D.

5) Is the increased INO80 enrichment in Figure 2C altered by ATMi?

6) In Figure 3B, it is difficult to compare interaction of WT and S412A ARP8 as the two data sets are on separate blots. Why does the interaction of WT become apparent only at 4 h, unlike at 2h in earlier blots?

7) Figure 1—figure supplement 1A – it is unclear what is being shown in Figure 1—figure supplement 1A. On the anti-Ino80 IP, the upper band appears to be phosphorylation in untreated conditions and the lower band appears to be a background band that copurifies with IgG. Which is the band we should be noticing? A molecular weight marker is needed on all gels using the ATM/ATR substrate antibody.

8) Figure 1C – why is the reduction of Arp8 phosphorylation not to the same degree as the phospho-Rpa2? Is the ATM/ATR substrate antibody detecting some other phospho substrate?

Similar comment for Figure 1—figure supplement 1E – even after treatment with siATM, there seems to be increased signal compared to the ctrl (-etoposide) sample. The reason for this needs to be resolved.

9) Figure 2 – while ChIP results are noted as statistically significant, the change in enrichment appears modest. What are the absolute values (not normalized) of enrichment? How does this (modest) fold change alter downstream genome stability? Similar questions apply to many of the ChIPs in this manuscript.

10) Figure 3B and Figure 3—figure supplement 1B – interaction of Ino80 appears modest. In Figure 3B it looks like it peaks at 2h then is reduced. Please explain why would interaction be reduced after 2 hours?

11) Figure 5 – the translocation results look clear, yet modest. Can the authors provide a point of reference to better judge these results? For example, what translocation rate would be expected upon overexpression or deletion of a repair protein?

12) The model of Arp8 phosphorylation weakening the interaction with Ino80 was a bit speculative, and the IP westerns not terribly convincing as is. We suggest probing for additional subunits in the IP westerns. I believe there are a couple of commercially available antibodies that can be used.

13) The authors should clearly discuss how ATM-mediated inhibition of INO80 and RAD51 loading at BCR serves to suppress translocations.

14) The authors should discuss if such a suppression of RAD51 loading is specific to BCR regions or is applicable to DSBs in general.

[Editors' note: further revisions were requested prior to acceptance, as described below.]

Thank you for resubmitting your work entitled "ATM regulates ARP8 phosphorylation to prevent chromosome translocations by counteracting DNA repair protein loading at damaged sites" for further consideration at *eLife*. Your revised article has been favorably evaluated by Jessica Tyler (Senior Editor), Jerry Workman (Reviewing Editor), and three reviewers.

The manuscript has been improved but there are some remaining issues that need to be addressed before acceptance, as outlined below:

Generally the authors have gone to great lengths to perform the requested experiments. Most are carried out adequately, although the results were not fully incorporated into the text and conclusions. In general, the findings are novel and important, as they provide relevant insight into the crosstalk of checkpoint kinases and chromatin remodelers in DNA repair and the phenotypes they monitor are indeed, cancer associated (chromosome translocation). Thus the significance of the paper is appropriate for *eLife*.

Major Comments:

1) New data provided in the revised manuscript suggests that a major kinase responsible for ARP8 phosphorylation after etoposide treatment is ATR (Figure 1E). ATM has a slight effect, but ATR (assuming the inhibitor used is specific) is more important. It is unclear still whether either kinase modifies directly or perhaps is upstream of another kinase. But given the fact that their own inhibitor study implicates ATR more strongly than ATM, the authors should test the effect of ATR inhibition on chromosome translocation. The effect might be more pronounced than ATM ablation. This is a minimal requirement for publication.

2) If ATR inhibition does not affect Rad51 loading then ATM is likely functioning locally at the translocation site, although ATR phosphorylates ARP8 globally. If acute ATR inhibition also influences chromosome translocation, authors should extensively re-write the manuscript so that both checkpoint kinases, ATM and ATR, are implicated in the phosphorylation of ARP8 and the prevention of chromosome translocation. The authors need to provide more detailed information on the conditions used for ATM and ATR inhibitor treatment. KU55933 and ATR inhibitor IV (=VE821, I believe) were shown to be highly selective but ATP competitive inhibitors are non-specific at the high dose.

3) Finally, based on the outcome of testing ATR knockdown or inhibition in some of the assays already included, the authors should change the text, main title and subtitles to recognize their own ATR results (some of which are already shown. The more accurate title is likely to be:

"Checkpoint kinases ATR and ATM regulate Arp8 phosphorylation and attenuate protein factor loading at double strand breaks."

---

## [Author Response]

Essential revisions:1) In the current manuscript, we did not find data that shows direct phosphorylation of ARP8 S412 by ATM, as the title of this manuscript indicates. The antibody probes for ATM or ATR targets and ATM inhibition only partially inhibits ARP8 phosphorylation detected by the antibody. Furthermore, ARP8 S412 phosphorylation status in ATM deficient BIVA cells compared to the ATM proficient cells was not shown. The onset of S412 phosphorylation after etoposide treatment is rather late. To correlate ATM action in this event, authors should compare the kinetics of ATM activation and another ATM target in parallel to ARP8 S412. As in the Discussion, it is plausible that ATR is also responsible for the ARP8 S412 phosphorylation. Phosphorylation analysis in combination with an ATR inhibitor would help to answer this question. To evaluate the phenotype that is influenced by ARP8 S412 phosphorylation and ATM, you should analyze the epistatic relation. Is S412A mutation additive or not to the ATM deficiency or ATM inhibition? For example, does ARP8 S412A mutation increase INO80 and RAD51 binding to the break point and 11q23 translocation increased in BIVA cells or by ATM inhibition in ATM proficient cells? Basically, we feel that there are many over-interpretations of these data throughout the manuscript (title, subtitles, figure legends etc.).Under the current data set, the authors should not state "ATM phosphorylates ARP8", rather than "ATM regulates ARP8 phosphorylation", otherwise authors should provide an in vitro ATM kinase assay on ARP8.

Thank you very much for the constructive comments to improve our manuscript. We agree with the reviewers that our findings are not sufficient to indicate the direct phosphorylation of ARP8 S 412 by ATM.

Following the reviewers’ advice, we compared the kinetics of ATM activation and other ATM targets, H2AX and RPA2, in parallel with ARP8 S412. As a result, the phosphorylations of ATM and H2AX were detected at 30 min after the induction of DNA damage. In contrast, the phosphorylations of RPA2 and ARP8 were relatively slow, and increased at 2 hrs and peaked at 4 hrs after etoposide treatment (Figure 1B). These findings suggest that ATM may activate other kinases to phosphorylate ARP8 after etoposide treatment. This notion was also supported by the results from an experiment to compare the etoposide-induced phosphorylation status of ARP8 in ATM-deficient BIVA cells and ATM inhibitor-treated ATM-proficient 11-4 cells (Figure 1D and E). The phosphorylation of ARP8 was not completely abolished in ATM-deficient and ATM inhibitor-treated cells.

Next, we examined the involvement of ATR in the phosphorylation of ARP8 after etoposide treatment, according to the reviewers’ suggestion. An ATR inhibitor also repressed the phosphorylation of ARP8 after etoposide treatment. This finding suggests the involvement of ATR in the regulation of ARP8 phosphorylation after etoposide treatment (Figure 1 E). Taken together, these results suggest that ATM and ATR regulate the phosphorylation of ARP8 after etoposide treatment.

Following the reviewers’ advice, we also examined the effect of ATM inhibition on the ARP8 S412A mutant by the ChIP analysis of 11-4 cells after etoposide treatment, using antibodies against INO80 and RAD51. We could not detect the additive effect of the ATM inhibitor on the ARP8 S412A mutation (Figure 2C–D, Figure 4B-C). ATMi treatment failed to enhance the etoposide-induced 11q23 translocations in ARP8 S412A mutant-expressing cells (Figure 5E). This finding suggests that ATM regulates ARP8 phosphorylation at S412, to regulate the loading of INO80 and RAD51 on BCR after etoposide treatment. In the revised manuscript, the original Figures 4A and 4B are presented as Figure 4—figure supplement 1C and Figure 4A, respectively. We have also replaced original Figure 4—figure supplement 1B with data from six independent experiments.

Taken together, these findings suggest that the etoposide-induced phosphorylation of ARP8 at S412 is regulated by ATM and ATR. Accordingly, we have changed the title to “ATM regulates ARP8 phosphorylation to prevent chromosome translocations by counteracting repair proteins loading at damaged sites”, according to the reviewers’ suggestions.

2) Data showing that ARP8 phosphorylation is attenuated by ATMi could be bolstered by examining phosphorylation status of ARP8 in ATM-deficient cells (ATVBIVA). A possible role of ATR in this phosphorylation should be investigated at least by using specific ATRi.

Following the reviewer’s comment, we examined the phosphorylation status of ARP8 in ATM-deficient BIVA cells after etoposide treatment (lanes 1-4, Figure 1D in the revised manuscript). As a result, the etoposide-induced phosphorylation of ARP8 was repressed in BIVA cells. Although the basal level of ARP8 phosphorylation in BIVA cells was higher than that in ATM-proficient 11-4 cells (Figure 1D), this finding confirms the involvement of ATM in the regulation of the etoposide-induced phosphorylation of ARP8.

We also examined the effect of ATRi on the phosphorylation status of ARP8 after etoposide treatment in ATM-proficient 11-4 cells (lanes 9-12, Figure 1E). We found that ATRi repressed the phosphorylation of ARP8 in ATM-proficient cells. These findings suggest that ATM and ATR regulate the etoposide-induced ARP8 phosphorylation. Accordingly, we have revised manuscript by including these findings.

3) Since gammaH2AX recruits INO80 to DSBs, the ChIP analyses in Figure 2 could be extended to gammaH2AX to confirm that etoposide induces DSBs specifically at the BCR.

Following the reviewer’s comment, we performed a chromatin immunoprecipitation assay using anti-gammaH2AX antibodies before and after etoposide treatment. The enrichment of gammaH2AX on BCR was observed after etoposide treatment, confirming that etoposide induces DNA damage specifically at the BCR. Accordingly, we have presented this result in the revised Figure 2—figure supplement 1A.

4) Does the S412D mutant fail to interact with INO80 in co-IPs? That could help explain the lack of INO80 enrichment in Figure 2D.

Following the reviewer’s suggestion, we performed the immunoprecipitation analysis to examine the interaction between S412D and INO80 in ATM-deficient cells (Figure 3—figure supplement 1B, in the revised manuscript). The interaction of the ARP8 S412D mutant with INO80 was weaker than that of ARP8 WT. The weaker interaction of the ARP8 S412D mutant with INO80 was also confirmed by the Proximity ligation assay (PLA) assay, as shown in Figure 3—figure supplement 2 in the revised manuscript. Accordingly, we have revised the manuscript by including these figures.

5) Is the increased INO80 enrichment in Figure 2C altered by ATMi?

We performed a chromatin immunoprecipitation analysis of INO80 using ATMi treated 11-4 Flp-in cells expressing either ARP8 wild-type or S412A mutant. The INO80 enrichment onto MLL BCR was increased by ATMi in ARP8 wild-type expressing cells, while no significant changes were observed in ARP8 S412A mutant expressing cells. This finding suggests the epistatic relation between ATM and ARP8 S412 phosphorylation. Accordingly, the data are presented in Figure 2C and D in the revised version.

6) In Figure 3B, it is difficult to compare interaction of WT and S412A ARP8 as the two data sets are on separate blots. Why does the interaction of WT become apparent only at 4 h, unlike at 2h in earlier blots?

Following the reviewer’s advice, we applied two sets of samples on one blot. The amounts of INO80 and HA-ARP8 from three independent experiments were quantified, using the Image J software. The S412A mutation increased the interaction of ARP8 with INO80. The gel images and quantitative results are presented in Figure 3B in the revised version.

7) Figure 1—figure supplement 1A – it is unclear what is being shown in Figure 1—figure supplement 1A. On the anti-Ino80 IP, the upper band appears to be phosphorylation in untreated conditions and the lower band appears to be a background band that copurifies with IgG. Which is the band we should be noticing? A molecular weight marker is needed on all gels using the ATM/ATR substrate antibody.

We apologize for the confusing data. We have added a molecular weight marker, and denoted the position of INO80 using an arrow in the revised Figure 1—figure supplement 1A. The upper bands detected by the anti-ATM/ATR substrate antibodies showed no significant difference after etoposide treatment. Since the size of the band is consistent with that of INO80, this upper band could be INO80 with background phosphorylation. The other possibility is that the upper band is non-specific. We agree with the reviewers that the lower band is a non-specific background band.

Accordingly, we have revised Figure 1—figure supplement 1A.

8) Figure 1C – why is the reduction of Arp8 phosphorylation not to the same degree as the phospho-Rpa2? Is the ATM/ATR substrate antibody detecting some other phospho substrate?Similar comment for Figure 1—figure supplement 1E – even after treatment with siATM, there seems to be increased signal compared to the ctrl (-etoposide) sample. The reason for this needs to be resolved.

Thank you very much for pointing out this important issue. There are two possible explanations for this issue. There could be a difference in the detection efficiency of the phosphorylated forms of ARP8 and RPA2. We used antibodies against ATM/ATR substrates for the detection of phosphorylated ARP8. In contrast, we applied antibodies against RPA2 to detect the phosphorylated form of endogenous RPA2, as slowly migrating bands. The other possibility is that ATR is more deeply involved in the regulation of ARP8 phosphorylation than that of RPA2 after etoposide treatment.

The phosphorylation of ARP8, as detected by the ATM/ATR substrate antibodies, was completely abolished by the S412A mutation. Therefore, the ATM/ATR substrate antibody should detect the phosphorylation of ARP8 at S412.

As the reviewer pointed out, the phosphorylated form of ARP8 was still detected in ATM-depleted cells after etoposide treatment in Figure 1—figure supplement 1E. Since we found that ATR is also responsible for the phosphorylation of ARP8 after etoposide treatment, from the experiments suggested by the reviewer (Figure 1E in the revised manuscript), ATR could facilitate phosphorylation of ARP8 even in ATM-depleted cells after etoposide treatment.

9) Figure 2 – while ChIP results are noted as statistically significant, the change in enrichment appears modest. What are the absolute values (not normalized) of enrichment? How does this (modest) fold change alter downstream genome stability? Similar questions apply to many of the ChIPs in this manuscript.

In the revised manuscript, we have added the absolute values of enrichment in the ChIP analysis, in Figure 2—source data 1.

As the reviewer pointed out, the change in enrichment of INO80 and RAD51 at BCR is modest, but increases chromosome translocations involving BCR. The possible explanation for this is that the binding of chromatin remodelers and DNA repair factors on damaged chromatin is dynamic, not static. Such a dynamic binding could reduce the ChIP value.

10) Figure 3B and Figure 3—figure supplement 1B – interaction of Ino80 appears modest. In Figure 3B it looks like it peaks at 2h then is reduced. Please explain why would interaction be reduced after 2 hours?

As in the original Figure 3B, the interaction of INO80 with the ARP8 S412A mutant also apparently peaks at 2h and then is reduced, in the revised Figure 3B. Factors associated with the homologous recombination repair pathway (HR) may be involved in the regulation of INO80 binding to ARP8. The other possibility is that the phosphorylation of ARP8 may facilitate the dissociation of the INO80 complex from damaged chromatin after the appropriate remodeling of damaged chromatin for DNA repair to avoid illegitimate recombination, leading to chromosome abnormalities. These possibilities are discussed in the second paragraph of the Discussion.

11) Figure 5 – the translocation results look clear, yet modest. Can the authors provide a point of reference to better judge these results? For example, what translocation rate would be expected upon overexpression or deletion of a repair protein?

Following the reviewer’s suggestion, we cited two papers here. One study showed the increased chromosome translocations in RAD51-overexpressing cells (Richardson, Stark, Ommundsen, and Jasin, 2004). The other paper from Mizutani’s lab showed the increased incidence of 11q23 translocations in ATM-deficient cells (Nakada et al., 2006). We have reported that the percentage of the ATM-deficient BIVA cells with the 11q23 chromosome splitting signal is increased from 0.17% to 2.5% by etoposide treatment, and in etoposide-treated ATM-proficient cells from 1.33% to 5.83% after ATMi treatment (Sun J. et al., 2010).

12) The model of Arp8 phosphorylation weakening the interaction with Ino80 was a bit speculative, and the IP westerns not terribly convincing as is. We suggest probing for additional subunits in the IP westerns. I believe there are a couple of commercially available antibodies that can be used.

I agree that the IP westerns are not convincing. Following the reviewer’s comment, we reprobed the blots with antibodies against ARP5. As a result, similar to INO80, we could observe a slightly increased amount of ARP5 in the S412A precipitated blot, as compared with that of wild-type ARP8 in ATM proficient cells. Similar results were obtained in wild-type ARP8, but not in S412D-expressing, ATM-deficient BIVA cells. Moreover, we also confirmed the role of the phosphorylation of ARP8 in the interaction with INO80 using a Proximity ligation assay, as described above. We would like to present the following data to the reviewers.

Examination of the interaction between INO80 and ARP8 in U2OS cells expressing HA-tagged wild-type or S412A ARP8. The endogenous ARP8-depleted cells were treated with etoposide for 15 min. After the cells were washed, they were placed in fresh medium and harvested at the indicated time points. The nuclear extracts were incubated with anti-HA-conjugated anti-mouse IgG Dynabeads. The precipitates were electrophoresed through a gel and probed by western blotting with an anti-INO80 or an anti-HA or an anti-ATM/ATR substrate antibody. The amounts of INO80 and HA-ARP8 were quantified, using the Image J software. The results of quantitative analysis are shown as the relative values as compared to the DMSO control from three independent experiments.

**Author response image 2. respfig2:** 

Co-immunoprecipitation analysis of ARP8 and INO80. ATM-deficient BIVA cells were co-transfected with the siARP8 and siARP8-resistant HA-tagged wild-type ARP8 or phospho-mimetic S412D ARP8 mutant. After etoposide removal, the cells were recovered at the indicated time points. The nuclear extracts were incubated with anti-HA-conjugated anti-mouse IgG Dynabeads. The precipitates were electrophoresed through a gel and probed by western blotting with either an anti-INO80 antibody or an anti-HA antibody. The amounts of INO80 and HA-ARP8 were quantified, using the Image J software. The results of quantitative analysis are shown as the relative values as compared to the DMSO control, from three independent experiments.

13) The authors should clearly discuss how ATM-mediated inhibition of INO80 and RAD51 loading at BCR serves to suppress translocations.

According to the reviewer’s comment, we have rewritten the second and third paragraphs of the Discussion, as follows:

“In response to DNA damage, the DNA repair process is facilitated by the phosphorylation of various proteins, including DNA repair factors, cell cycle regulators, and chromatin relaxing factors, through the positive regulation by ATM and ATR ((Marechal and Zou, 2013) (Cimprich and Cortez, 2008) Clouaire et al., 2017; Shiloh, 2003; Shiloh and Ziv, 2013). […] The phosphorylation of ARP8 after etoposide treatment could be regulated by multiple steps and factors, for the precise control of DNA repair activity to maintain chromosome stability.”

14) The authors should discuss if such a suppression of RAD51 loading is specific to BCR regions or is applicable to DSBs in general.

According to the reviewer’s comment, we have added the following sentences at the end of the fourth paragraph of the Discussion:

“INO80 and ARP8 have been shown to regulate the RAD51 loading to damaged chromatin in yeast (Tsukuda et al., 2005) (Tsukuda et al., 2009) (van Attikum, Fritsch, and Gasser, 2007) (Lademann et al., 2017). […] Therefore, this regulation of HR by ARP8 in human cells may also be applicable to DSBs in general.”

[Editors' note: further revisions were requested prior to acceptance, as described below.]

Major Comments:1) New data provided in the revised manuscript suggests that a major kinase responsible for ARP8 phosphorylation after etoposide treatment is ATR (Figure 1E). ATM has a slight effect, but ATR (assuming the inhibitor used is specific) is more important. It is unclear still whether either kinase modifies directly or perhaps is upstream of another kinase. But given the fact that their own inhibitor study implicates ATR more strongly than ATM, the authors should test the effect of ATR inhibition on chromosome translocation. The effect might be more pronounced than ATM ablation. This is a minimal requirement for publication.

Thanks for the important comment. Following the editor’s advice, we examined the effect of the ATR inhibitor VE821 on the etoposide-induced 11q23 chromosome translocations in ATM-proficient 11-4 cells, by the dual color FISH analysis using the MLL gene probes (Figure 6B). The FISH analysis revealed that the increase of the split signal positive cells by ATRi was less than that by ATMi. Although ATR is suggested to be the major kinase responsible for ARP8 phosphorylation after etoposide treatment, this finding suggests that the effect of ATRi on the etoposide-induced chromosome translocations is limited. Moreover, no additional effects of ATRi on the increase of the chromosome translocations by ATMi were observed. Taken together, these findings strongly suggest that ATM is likely functioning locally at the translocation site, while ATR phosphorylates ARP8 globally. Accordingly, we have added a section “ATM, but not ATR, negatively regulates RAD51 loading onto the BCR after etoposide treatment to repress 11q23 chromosome translocations” in the “Results”.

2) If ATR inhibition does not affect Rad51 loading then ATM is likely functioning locally at the translocation site, although ATR phosphorylates ARP8 globally. If acute ATR inhibition also influences chromosome translocation, authors should extensively re-write the manuscript so that both checkpoint kinases, ATM and ATR, are implicated in the phosphorylation of ARP8 and the prevention of chromosome translocation. The authors need to provide more detailed information on the conditions used for ATM and ATR inhibitor treatment. KU55933 and ATR inhibitor IV (=VE821, I believe) were shown to be highly selective but ATP competitive inhibitors are non-specific at the high dose.

We examined the effect of ATR inhibition on the RAD51 loading on the BCR of MLL in 11-4 cells after etoposide treatment, by a ChIP analysis. In contrast to the increased loading of RAD51 onto the BCR of MLL after etoposide treatment by ATMi, ATRi failed to do so (Figure 6B). This finding is consistent with the results obtained by the FISH experiments described above. As the editors suggested, ATM is likely functioning locally at the translocation sites, while ATR is functioning globally. Accordingly, we have discussed this point in the fourth paragraph in the “Discussion”.

We apologize for providing insufficient information about the conditions used for the ATM and ATR inhibitor treatments. We now provide detailed information in the “Materials and methods” and the figure legends in the revised manuscript. As the reviewer speculated, the ATR inhibitor we used was VE821.

3) Finally, based on the outcome of testing ATR knockdown or inhibition in some of the assays already included, the authors should change the text, main title and subtitles to recognize their own ATR results (some of which are already shown. The more accurate title is likely to be:"Checkpoint kinases ATR and ATM regulate Arp8 phosphorylation and attenuate protein factor loading at double strand breaks."

The FISH analysis and the chromatin immunoprecipitation experiments suggested by the editors showed that ATR inhibition did not change the RAD51 loading onto the BCR of MLL after etoposide treatment, while a slight increase in 11q23 chromosome translocation split signal cells was detected (Figure 6). These results suggest that the ARP8 phosphorylation regulated by ATR is functionally distinct from that by ATM. Accordingly, we have changed the title to “Distinct roles of ATM and ATR in the regulation of ARP8 phosphorylation to prevent chromosome translocations”.